# 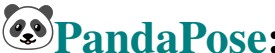PandaPose:
# 3D Human Pose Lifting from a Single Image via Propagating 2D Pose Prior to 3D Anchor Space

**Jinghong Zheng[1]   Changlong Jiang[1]   Yang Xiao[1,†]   Jiaqi Li [2]**

**Haohong Kuang[1]   Hang Xu[1]   Ran Wang[3,4]   Zhiguo Cao[2]   Min Du [5]   Joey Tianyi Zhou [6,7]**

[1] National Key Laboratory of Multispectral Information Intelligent Processing Technology, School of Artificial Intelligence and Automation, Huazhong University of Science and Technology, Wuhan 430074, China

[2]School of Artificial Intelligence and Automation, Huazhong University of Science and Technology

[3]School of Journalism and Information Communication, Huazhong University of Science and Technology

[4]School of Future Technology, Huazhong University of Science and Technology

[5] ByteDance Inc.

[6] Centre for Frontier AI Research, Agency for Science, Technology and Research, Singapore

[7] Institute of High Performance Computing, Agency for Science, Technology and Research, Singapore

[†]Corresponding author

{deepzheng,changlongj,Yang_Xiao,lijiaqi_mail}@hust.edu.cn
{haohong_kuang,hang_xu,rex_wang,zgcao}@hust.edu.cn
bingwen.ai@bytedance.com
joey_zhou@a-star.edu.sg

## Abstract

3D human pose lifting from a single RGB image is a challenging task in 3D vision. Existing methods typically establish a direct joint-to-joint mapping from 2D to 3D poses based on 2D features. This formulation suffers from two fundamental limitations: inevitable error propagation from input predicted 2D pose to 3D predictions and inherent difficulties in handling self-occlusion cases. In this paper, we propose PandaPose, a 3D human pose lifting approach via propagating 2D pose prior to 3D anchor space as the unified intermediate representation. Specifically, our 3D anchor space comprises: (1) Joint-wise 3D anchors in the canonical coordinate system, providing accurate and robust priors to mitigate 2D pose estimation inaccuracies. (2) Depth-aware joint-wise feature lifting that hierarchically integrates depth information to resolve self-occlusion ambiguities. (3) The anchor-feature interaction decoder that incorporates 3D anchors with lifted features to generate unified anchor queries encapsulating joint-wise 3D anchor set, visual cues and geometric depth information. The anchor queries are further employed to facilitate anchor-to-joint ensemble prediction. Experiments on three well-established benchmarks (*i.e.*, Human3.6M, MPI-INF-3DHP and 3DPW) demonstrate the superiority of our proposition. The substantial reduction in error by 14.7% compared to SOTA methods on the challenging conditions of Human3.6M and qualitative comparisons further showcase the effectiveness and robustness of our approach.

39th Conference on Neural Information Processing Systems (NeurIPS 2025).

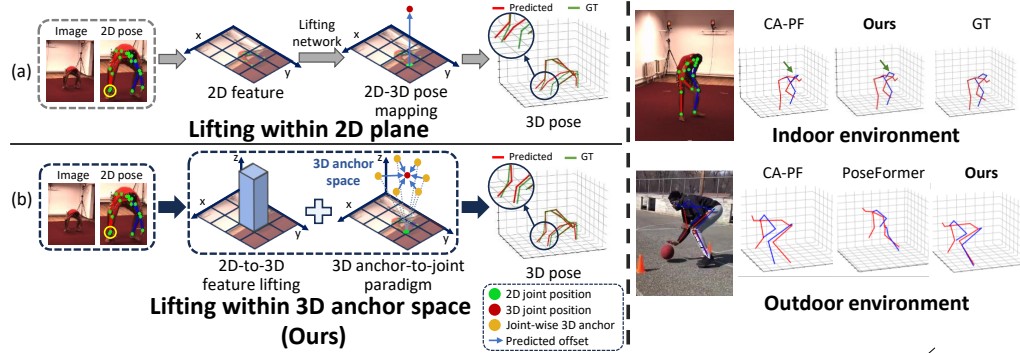

Figure 1: Comparison between different 2D-to-3D human pose lifting manners. Previous methods (a) generally concern 2D in-plane feature and directly predict 3D pose. Our method (b) mitigates depth ambiguity by lifting in-plane feature to 3D space and interacting with joint-wise 3D anchors. Then 3D pose will be estimated via anchor-to-joint ensemble prediction within 3D anchor space. Our method is robust to both indoor and outdoor occlusion scenarios.

# 1   Introduction

Monocular 3D human pose estimation from a single RGB image has wide applications including action recognition [44, 60, 5], virtual reality [11] and human-computer interaction [16, 34]. Compared to sequence based methods [27, 57, 20, 50], image based counterparts [48, 54, 55] generally have higher potential for real-time applications and insensitive to the length of input sequence. Recently, benefiting from the advancement in 2D human pose estimation [4, 33], the research of 3D pose lifting with input predicted 2D pose has drawn researchers' attention. However, many challenges remain in this field, including estimated 2D pose error propagation, high-frequency self-occlusion, *etc.* [53, 15].

Currently, the state-of-the-art image-based approaches [54, 55, 59] attempt to perform pose lifting by introducing image features to supplement spatial context as shown in Fig. 1(a). Although these methods demonstrate promising performance, they exhibit inherent limitations. First, they attempt to establish the one-to-one mapping from 2D to 3D pose, increasing the dependency on the 2D pose prediction quality. Consequently, minor noise in the input poses could lead to significant deviations of predicted 3D pose, resulting in limited robustness. Second, self-occlusion of human body on images is quite common due to the monocular nature. Existing methods [55, 50, 28] primarily rely on image descriptive clues for 3D pose characterization, failing to explicitly model depth dimension and consequently struggling with depth ambiguity and self-occlusion challenges. To overcome these limitations, we propose to *facilitate 3D pose lifting via propagating 2D pose prior to 3D anchor space*. Therefore, joint-wise 3D anchor setting and depth-aware feature lifting are proposed for providing a robust anchor initialization to suppress noise and integrating hierarchical depth to eliminate ambiguities in self-occlusion, forming the 3D anchor space, as shown in Fig. 1(b).

For joint-wise 3D anchors, we aim to propagate the input 2D pose priors into 3D joint priors that possess both robust error tolerance and high accuracy. This mitigates error propagation issue in previous pose lifting methods, which overly rely on the accuracy of input poses. Specifically, we utilize 3D anchor set as a coarse initialization relevant to input poses and predict 3D anchor-to-joint offsets for each joint in an ensemble manner, rather than directly estimating 3D joint positions. The preliminary exploration in prior works [42, 14] are limited to a simplistic global fixed anchor setting, which often results in excessively long anchor-to-joint regression offsets, leading to degradation in accuracy and robustness with insufficient exploitation of the input pose. In contrast, we propose joint-wise local anchors that fully exploit the guidance of the global 2D pose context. By adaptively setting a cluster of anchors near each joint based on its 2D position in a learnable manner, aiming to achieve a trade-off between robustness to errors in the input and accuracy of the initial 3D priors.

Self-occlusion remains a significant challenge for current pose lifting methods, given the monocular nature of inputs images and poses. Solely rely on in-plane features can be severely semantically corrupted in occluded regions. Incorporating spatial structural information serves as a potential solution to recover predictions for self-occluded regions in other domains [17, 6, 51]. However, simply introducing depth maps proves inadequate for millimeter-level human pose estimation, particularly failing to resolve depth ambiguity in self-occluded joints. Consequently, we propose a joint-wise

feature lifting module that aligns image features to the domain of 3D anchor features under the guidance of intermediate joint-level depth supervision. For the supervision, in the absence of dense ground truth depth maps, we extract the depth value of ground truth joints instead of the complete depth as supervision. This approach not only inherently furnishes crucial prerequisites for the reconstruction of self-occluded joints through depth stratification but also facilitates a more practical approach to fitting the actual joint depth distributions. At the feature level, we leverage 2D pose priors for feature sampling of visual features to resist background noise, simultaneously reducing computational memory cost. The sampled features interact with depth information before being projected into 3D anchor space, enabling robust joint localization against occlusions.

The initial anchor queries, derived from joint-wise 3D anchor setting, undergo multiple attention-based interactions with depth features and lifted image features within the 3D anchor space. The resulting output queries integrate cross-modal information from visual, depth, and geometric anchor data, which are then transformed into anchor offsets and weights towards each joint. This anchor-to-joint prediction mechanism produces robust 3D pose predictions that maintain both resilience to input 2D pose inaccuracies and effectiveness against self-occlusion.

The superiority of our proposed PandaPose is verified on three well-established datasets (*i.e.*, Human3.6M [13], MPI-INF-3DHP [26], 3DPW [39]). The experiments demonstrate our approach essentially outperforms all the state-of-the-art image based counterparts. Especially under challenging scenarios (*e.g.* occlusion) in Human3.6M, our method achieves a significant improvement of 11.3% in MPJPE and 14.7% in PA-MPJPE. Overall, the main contributions of this paper include:

- We propose PandaPose to address image based 3D human pose lifting via propagating 2D pose prior to 3D anchor space as intermediate representation, and achieve effective and occlusion-resistant 3D pose estimation through anchor-to-joint ensemble prediction;
- We design joint-wise 3D anchor setting to provide an accurate and robust mapping to 3D joint, thereby mitigating the impact of input 2D pose inaccuracies;
- A novel 2D-to-3D feature lifting method is proposed to resist self-occlusion and depth ambiguity issues, via estimating joint-wise depth distribution with sparse depth supervision.

## 2 Related Works

**2D-to-3D human pose lifting.** With advancements in 2D human pose estimation [4, 33], lifting 2D poses to 3D has become a critical research area, with methods generally categorized into sequence-based and image-based approaches. Sequence-based methods [27, 22, 40, 47, 57, 19, 20, 50, 31, 28] use long temporal sequences using GCN [52] or Transformer [38] to model spatial-temporal correlations in 2D poses. Despite significant performance gains with longer sequences [50, 20], these methods also face increased computational complexity and memory demands. With single frame input, some works [43, 48, 56, 21] attempt to model the spatial relationships within the human skeleton to capture the spatial correlation between 2D and 3D poses. Other approaches propose to leverage visual features [54, 55, 59] from 2D pose estimators to compensate for the weak descriptive capability of 2D pose positions. However, due to the limitations of in-plane features, they commonly lack explicit modeling of depth, leading to challenges in handling depth ambiguity and self-occlusion. In our work, we explicitly incorporate joint-wise depth distribution to enhance feature with depth awareness and alleviate prediction difficulty using an adaptive anchor-to-joint regression manner.

**Anchor based pose estimation.** In hand pose estimation, some works [42, 12, 8] designed anchor-to-joint paradigm by treating 2D anchors as local regressors to estimate hand poses. A2J-Transformer [14] further proposed combining the 3D anchor setting with Transformer [23] to enhance the resistance towards occlusion. However, the anchor settings in these methods are fixed and cannot adapt to the content of the image, leading to suboptimal performance in certain scenarios. We are the first to explore the use of anchors in 3D human pose lifting and leverage 2D pose prior to enhance anchor design from static to learnable adaptive.

**Visual feature with 3D enhancement.** For pose estimation, some works expanded visual features into 3D voxels [37, 45, 32, 25] and applied 3D CNNs to regress 3D human poses. Although effective, the accuracy of 3D human pose estimation may be constrained by the voxel resolution due to limitations in memory usage and computational cost. To efficiently construct 3D feature representations, some works in 3D object recognition [29, 17, 51, 6, 46] have made positive explorations. These methods

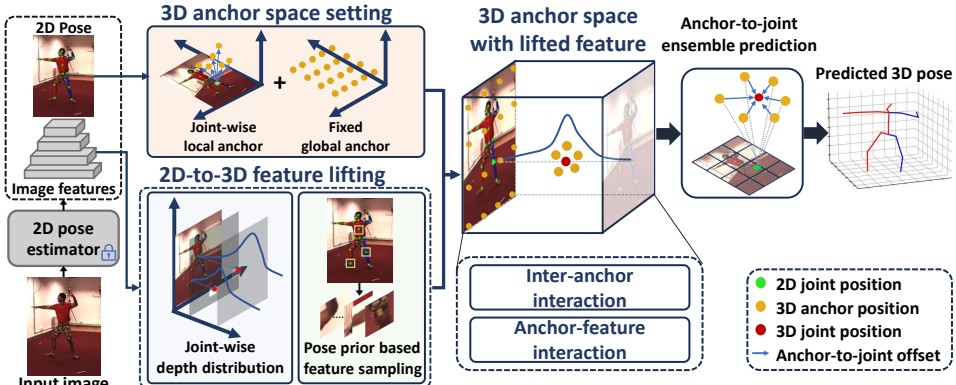

Figure 2: Overview of PandaPose pipeline. Given input single-frame 2D pose and intermediate image features, we adaptively sample anchors in 3D space. By estimating joint-wise depth distributions and employing a 2D pose prior based sampling strategy, we lift features from 2D to 3D domain. After 3D anchor-feature interaction, we obtain predicted 3D pose through anchor-to-joint ensemble prediction.

typically extend in-plane features into 3D space using given or predicted depth maps. However, a single foreground depth map is insufficient for accurate 3D pose estimation, especially under self-occlusion with significant depth ambiguity. Therefore, we propose to predict joint-wise depth distributions and integrate them with in-plane features to lift them into depth-aware 3D features.

## 3 Method

### 3.1 Overview of main pipeline

The main technical pipeline is shown in Fig. 2. Following the two-stage pipeline of SOTA pose lifting methods [55], given the input RGB image $I$ of size $H \times W \times 3$, an off-the-shelf 2D pose estimator [33] generates the corresponding 2D human pose $P_J^{2D} \in \mathbb{R}^{N_J \times 2}$ along with intermediate pyramid feature maps, where $N_J$ is the number of joints. Given the input 2D pose, we propose to form 3D anchor set $A$ via adaptive joint-wise local anchor and fixed global anchor setting (Sec. 3.2). To lift the input visual 2D in-plane features into depth-aware 3D features, we predict the joint-wise depth distribution maps $Dist_D$ and depth embedding $F_D$ (Sec. 3.3). We also design a 2D pose prior based feature sampling strategy to extract joint-related visual features $F_I$ (Sec. 3.4), aiming to minimize computational costs while filtering out background noise. The 3D anchors are encoded as learnable anchor queries $Q_{anchor}$ and interact with lifted 3D features within 3D anchor space, thus making anchor queries as an unified representation encapsulating 3D anchors, visual cues and depth information (Sec. 3.5). Finally, the predicted 3D pose $P_J^{3D} \in \mathbb{R}^{N_J \times 3}$ is obtained through anchor-to-joint ensemble prediction (Sec. 3.6).

### 3.2 Joint-wise adaptive 3D anchor setting

In contrast to current pose lifting methods that typically establish a direct joint-to-joint mapping from 2D to 3D poses, we construct 3D anchors as coarse initialization relevant to the input 2D poses and then predict 3D anchor-to-joint offsets for each joint in an ensemble manner. Previous anchor-based methods [42, 14] typically employ fixed anchor settings sparsely located in 3D space, lacking adaptability to specific pose patterns and thus leading to large offsets for distant anchors from human regions (Fig. 3). Given that predicting large offsets involves a greater margin for error, it potentially degrades overall performance in joint localization, especially under occlusions or 2D pose inaccuracies. To address this, we propose evolving the anchor setting from static to dynamic adaptive by leveraging 2D pose priors. We conduct a comparison with different anchor settings on the Human3.6M [13] (Fig. 3). By selecting top-50 anchor-to-joint weights, we identify informative anchors contributing most to joint prediction and calculate their proportion relative to the total. Our adaptive setting significantly reduces the offset from 154.6$mm$ to 69.7$mm$.

The adaptive 3D anchor generation procedure is shown in Fig. 4. The input 2D pose $P_J^{2D}$ is first normalized to $[-1, 1]$. Then 3D local anchors are adaptively generated, producing a set of $K$ sampled 3D offsets $\delta \in \mathbb{R}^{K \times 3}$ for each joint $j \in J$:

$$\delta_J = Linear(P_J^{2D}), \quad \delta_J \in \mathbb{R}^{N_J \times K \times 3}. \tag{1}$$

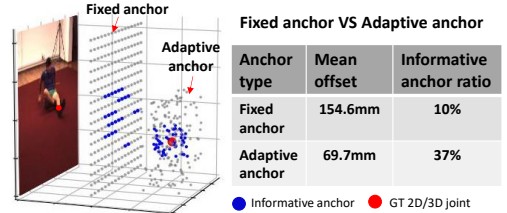

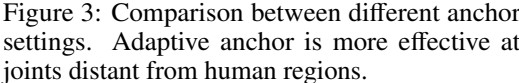

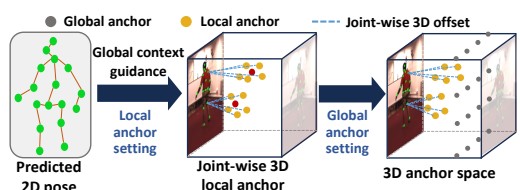

Figure 3: Comparison between different anchor settings. Adaptive anchor is more effective at joints distant from human regions.

Figure 4: Illustration of adaptive 3D anchor setting including joint-wise local anchor and global fixed anchor.

Due to the consideration of global 2D pose context, it has a good adaptability to locally inaccurate 2D joints, as shown in Fig. 10. Simultaneously, the normalized 2D joint position $(j_x, j_y)$ is initialized as a 3D position at depth 0, formed $(j_x, j_y, 0)$. The 3D local anchor set $A_{local}$ is then generated by adding the sampling offsets to the corresponding joint 3D position $(j_x, j_y, 0)$:

$$A_{local} = \left\{ a \mid P_a = (j_x, j_y, 0) + \delta_{j,k}, j \in J, k \in K \right\}. \tag{2}$$

To enhance the model's generalizability and training stability, we integrate a subset of global fixed anchors to complement local adaptive anchors with a stable global context. Specifically, we preset 3D global anchors $A_{global} \in \mathbb{R}^{256 \times 3}$, which are uniformly distributed on the plane of root joint in 3D space with in-plane stride $S_x = H/16$ and $S_y = W/16$. Finally, the 3D anchor set $A$ is the combination of global anchors and local anchors:

$$A = A_{global} \cup A_{local}. \tag{3}$$

The 3D anchor set $A$ are then used to facilitate anchor-feature interaction in the form of anchor query $Q_{anchor}$. Their 3D position $P_A^{3D}$ is the initial position for anchor-to-joint prediction.

### 3.3 Joint-wise depth distribution estimation

In 3D human pose lifting, one of the key challenges lies in resolving self-occlusion. Current methods are generally limited by relying on constrained 2D features, which leads to difficulties in handling depth ambiguity. However, simply estimating a single depth map as a supplement to image features is insufficient for achieving precise pose estimation requirements. First, pose estimation focuses more on the relative depth relationships between joints, and a single depth map cannot provide effective depth predictions for occluded or 2D-proximity joints. Second, without ground truth depth maps, how to enable models to perform more accurate depth estimations remains a challenge. As illustrated in Fig. 5, joints far apart in 3D space to appear close in 2D planes due to projection distortion, often overlapping in the downsampled feature map. This leads to ambiguous training, where one point maps to multiple GT depths, and inaccurate inference, where multiple joints share the same incorrect depth. Thus, the inherent depth ambiguities cannot be truly well resolved.

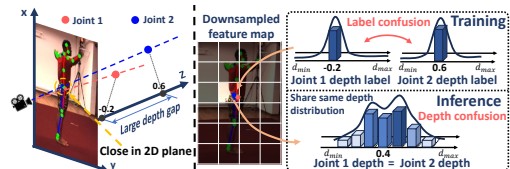

Figure 5: Existing methods typically predict a single depth map. Due to projection distortion, joints far apart in 3D space may appear close in 2D planes, often overlapping in the downsampled feature map, causing significant confusion during training and inference.

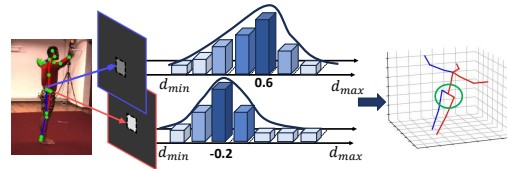

Figure 6: Illustration of joint-wise depth distribution. Each joint has a corresponding predicted depth map, allowing occluded or nearby joints on the plane to have distinct depth distributions.

Therefore, we innovatively design a joint-wise approach to predict individual depth distributions for each joint as shown in Fig. 6. The process of joint-wise depth distribution estimation is shown in Fig. 7(a). We utilize a light-weight depth net to estimate joint-wise depth (architecture is provided in Appendix A.2). We use multi-scale features from a pretrained 2D pose estimator to estimate depth distribution maps at corresponding scales. To balance efficiency with feature richness, we choose a single $H/8 \times W/8$ resolution feature map as input and estimate depth distributions for each

joint independently. To simplify the continuous depth value prediction, we segment the depth range $[-d_{min}, d_{max}]$ into $K_{bin}$ bins, treating each bin as a distinct class, thereby formulating depth estimation as a classification task [30, 1]. The resulting depth distribution maps $N_J \times H/8 \times W/8 \times K_{bin}$ are then upscaled via interpolation to match different image resolutions, forming $Dist_D$. Additionally, we use a single Transformer encoder layer [7] on the features output by depth net to generate depth-aware embeddings $F_D$, encoding geometric depth cues for 3D anchors.

To address the challenge of generating accurate depth distribution consistent with 3D pose in the absence of the ground-truth dense depth map, we introduce leveraging 3D pose annotations as sparse supervision. Consequently, we first map the depth value of GT 3D pose to the corresponding bin as the label, and then compute binary cross entropy loss [6] separately for each joint depth map. To mitigate learning complexity and filter out irrelevant noise, the loss calculation is limited to the $r \times r$ region surrounding the 2D joint.

$$\mathcal{L}_{depth} = \frac{1}{N} \sum_{n=1}^{N} \left( \sum_{k=0}^{l_n-1} log P^k_{(n,0)} + \sum_{k=l_n}^{K_{bin}} log P^k_{(n,1)} \right) \quad (4)$$

where $N = N_J \times r \times r$ indicates the total number of joint-wise depth supervision pixels.

### 3.4   2D pose prior based feature sampling

For 3D human pose estimation, attention-based models typically process each pixel, which can introduce background noise and increase computational load without improving accuracy. To address these, we propose a 2D pose prior based feature sampling strategy to improve the feature extraction of traditional attention modules. Our method leverages 2D poses as prior, selectively focusing on features within a $r \times r$ region around joints instead of the whole image (see Fig. 7(b)). This approach reduces irrelevant information interference while enhancing computational efficiency. Regarding multi-scale features, our strategy is applied to the first two high-resolution layers, with the sampling radius proportional to the resolution ($r = H/16$), preserving global semantic information in lower-resolution layers. And we normalize image feature channels to the same $C_I$. The similar sampling strategy is employed for depth distributions, ensuring that the $N_F$ tokens fed into the anchor feature interaction decoder are pixel-aligned, where $N_F$ is the number of pixel tokens that are summed after flattening the multi-layer sampled feature map. The sampled image features $F_I \in \mathbb{R}^{N_F \times C_I}$ and depth distributions $Dist_D \in \mathbb{R}^{N_F \times K_{bin}}$ are then processed with the anchor-feature interaction decoder for further interaction within 3D anchor space, enabling robust joint localization against occlusions.

### 3.5   3D anchor feature interaction

Within 3D anchor space, we propose to leverage learnable adaptive 3D anchors that interact with both depth and visual features to enhance spatial context understanding, addressing the challenges of accurately capturing spatial relationships and depth information in pose estimation. Structure of anchor feature interaction decoder is shown in Fig. 8. Based on learnable adaptive 3D anchors, we encode them as learnable anchor queries $Q_{anchor}$ to predict 3D pose through anchor-feature interaction decoder. Each decoder layer comprises a depth cross-attention layer, an inter-anchor self-attention layer, and a 3D deformable cross-attention layer. The depth cross-attention layer captures latent depth features, enabling anchors to adaptively understand spatial contexts from depth-guided regions, enhancing the perception of inter-joint depth relationships. In this process, we linearly transform the anchor query and depth embedding into the query $Q_D$, key $K_D$, and value $V_D$:

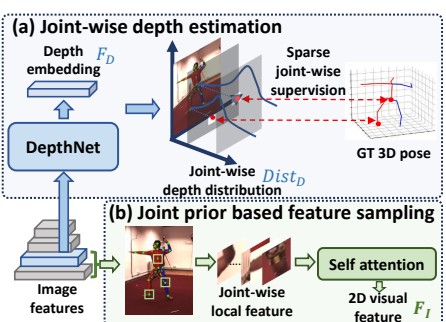

Figure 7: Processes of joint-wise depth distribution estimation and 2D pose prior based feature sampling.

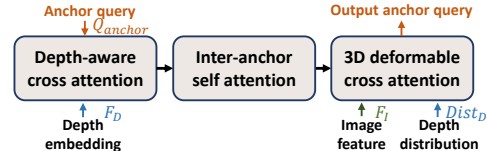

Figure 8: Structure of 3D anchor feature interaction decoder.

$$Q_D = Linear(Q_{anchor}), \quad K_D, V_D = Linear(F_D), \quad (5)$$

| | Method | Venue | Frame | Parameters (M) for Lifting Module | MPJPE ↓ | PA-MPJPE ↓ |
|---|---|---|---|---|---|---|
| Sequence based | PoseFormer [57] | ICCV'21 | 81 | 9.5 | 44.3 | 34.6 |
| | MHFormer [20] | CVPR'22 | 351 | 24.8 | 43.0 | 34.4 |
| | MixSTE [50] | CVPR'22 | 243 | 33.6 | 40.9 | 32.6 |
| | P-STMO [31] | ECCV'22 | 243 | 4.6 | 43.0 | 34.4 |
| | STCFormer [35] | CVPR'23 | 243 | 18.9 | 41.0 | 32.0 |
| | KTPFormer [28] | CVPR'24 | 243 | 35.2 | 40.1 | 31.9 |
| Image based | *Full test set* | | | | | |
| | GraphSH [43] | CVPR'21 | 1 | 3.7 | 51.9 | - |
| | HCSF [48] | ICCV'21 | 1 | - | 47.9 | 39.0 |
| | GraFormer [56] | CVPR'22 | 1 | - | 51.8 | - |
| | Diffpose [10] | CVPR'23 | 1 | 1.9 | 49.7 | - |
| | Zhou et al.[59] | AAAI'24 | 1 | - | 46.4 | - |
| | HiPART [58] | CVPR'25 | 1 | 2.4 | 42.0 | - |
| | CA-PF [55] | NeurIPS'23 | 1 | 14.1 | 41.4 | 33.5 |
| | PandaPose (Ours) | | 1 | 15.2 | **39.8** (1.6↓) | **32.7** (0.8↓) |
| | *Challenging subset* | | | | | |
| | CA-PF [55] | NeurIPS'23 | 1 | 14.1 | 82.4 | 82.0 |
| | PandaPose (Ours) | | 1 | 15.2 | **73.1** (9.3↓) | **69.9** (12.1↓) |

Table 1: Comparison with state-of-the-arts methods on Human3.6M. MPJPE and PA-MPJPE are reported in millimeters. The best results are shown in **bold**. Our method not only achieves leading performance on the full test set, but also shows significant improvement on challenging subset.

Then, the depth-aware 3D anchor queries are fed into the inter-anchor self-attention layer to promote articulated clues between anchors. Finally, we lift the flatten in-plane features $F_I$ into 3D space using the joint-wise depth distribution $Dist_D$ via the outer product $F_{3D} = Dist_D \otimes F_I$, and apply a 3D deformable cross-attention (DCA) layer to enable 3D anchor queries effectively aggregate visual characteristics of the 3D scene. Then, using the 3D anchors as reference points, for a specific anchor $a$ located at the position $[P_a]$, we perform feature interaction through 3D deformable cross-attention:

$$DCA(a) = \sum_{n \in N} W_n \phi(F_{3D}, P_a + \Delta S_n). \tag{6}$$

Following [61], each anchor is associated with $N$ adaptive sampling points near $P_A$, whose offsets $\Delta S_n$ are predicted from the input anchor query $Q_{anchor}$ through a linear layer. This allows the Transformer to dynamically attend to sparse yet semantically rich local regions in a data-driven manner, enhancing both efficiency and accuracy. The term $\phi(F_{3D}, P_a + \Delta S_n)$ denotes the trilinear interpolation to sample features from the expanded 3D feature map $F_{3D}$. The output anchor query forms a unified representation in the 3D anchor space, integrating multimodal information and helping to solve the depth ambiguity issues.

## 3.6 Anchor-to-joint prediction

With anchor queries $Q_{anchor}$ output from decoder, we use MLP layers to extract the offsets $O$ and weights $W$ of the anchors with respect to all joints. The joint positions are positioned in the form of weighted sum of anchor-to-joint offsets:

$$P_j^{3D} = \sum_{a \in A} \tilde{W}_{a,j}(P_a + O_{a,j}), \tag{7}$$

where $P_j$ and $P_a$ indicate the 3D position of the certain joint $j$ and anchor $a$. $O_{a,j}$ denotes the offset from $a$ towards $j$. $\tilde{W}_{a,j}$ is the softmax-derived weight of $a$ towards $j$.

## 3.7 Loss Function

We train the joint-wise depth distribution map in a sparsely supervised manner (Sec. 3.3), denoted as $\mathcal{L}_{depth}$, and use MPJPE [27, 57] to supervise the training of 3D pose, denoted as $\mathcal{L}_{pose}$. The overall loss function is formulated as:

$$\mathcal{L} = \lambda_1 \mathcal{L}_{pose} + \lambda_2 \mathcal{L}_{depth}. \tag{8}$$

Here we set $\lambda_1 = 2$ and $\lambda_2 = 0.1$ for scale balance.

## 4 Experiments

### 4.1 Datasets and evaluation metrics

| Method | PCK ↑ | AUC ↑ | MPJPE ↓ |
|---|---|---|---|
| *Full test set* | | | |
| GraFormer [56] | 79.0 | 43.8 | - |
| Li *et al.*[18] | 81.2 | 46.1 | 99.7 |
| HCSF [48] | 82.1 | 46.2 | - |
| Zhou *et al.*[59] | 88.2 | 59.3 | - |
| CA-PF [55] | 98.0 | 75.4 | 32.7 |
| PandaPose (Ours) | **98.6** (0.6↑) | **75.8** (0.4↑) | **31.8** (0.9↓) |
| *Challenging subset* | | | |
| CA-PF [55] | 84.5 | 53.2 | 66.6 |
| PandaPose (Ours) | **94.3** (9.8↑) | **62.5** (9.3↑) | **51.8** (14.8↓) |

Table 2: MPI-INF-3DHP comparisons with image-based methods.

| Method | MPJPE | PA-MPJPE ↓ |
|---|---|---|
| STRGCN [2] | 112.9 | 69.6 |
| VideoPose [27] | 101.8 | 63.0 |
| PoseFormer [57] | 118.2 | 73.1 |
| Learning [49] | 91.1 | 54.3 |
| PCT [9] | 83.1 | 53.9 |
| DiffPose[10] | 82.7 | 53.8 |
| Di$^2$Pose [41] | 79.3 | 50.1 |
| HiPART [58] | 77.2 | 48.8 |
| PandaPose (Ours) | **74.9**(2.3↓) | **46.9**(1.9↓) |

Table 3: Cross-dataset comparison of our method with SOTA methods on 3DPW.

| Global fixed anchor | Adaptive local anchor | MPJPE↓ (Full) | MPJPE ↓ (Challenging) |
|---|---|---|---|
| PandaPose w/o anchor | | 42.1 | 81.9 |
| ✓ | | 40.8 (1.3↓) | 76.2 (5.0↓) |
| | ✓ | 40.1 (2.1↓) | 74.0 (7.2↓) |
| ✓ | ✓ | 39.8 (2.3↓) | 73.1 (8.1↓) |

Table 4: Anchor setting strategy comparison.

| Anchor feature | Depth distribution | MPJPE↓ (Full) | MPJPE ↓ (Challenging) |
|---|---|---|---|
| 2D | - | 40.9 | 80.8 |
| 3D | Single | 40.3 (0.6↓) | 75.9 (4.9↓) |
| 3D | Joint-wise | 39.8 (1.1↓) | 73.1 (7.7↓) |

Table 5: Ablation study of joint-wise 3D feature lifting.

We train our model separately on the two commonly used 3D human pose datasets (*i.e.*, **Human3.6M** [13] and **MPI-INF-3DHP** [26]) to demonstrate the effectiveness of PandaPose. To better verify the generalization ability, we train our model on Human3.6M and conduct a cross-dataset evaluation on **3DPW** [39] in-the-wild dataset. To better illustrate the robust to occlusion and 2D pose inaccuracy, we select samples from Human3.6M and MPI-INF-3DHP with an error > 5 between the predicted 2D pose and GT 2D pose as the *challenging subset* ($\approx 5\%$ in dataset). Additional general information about each dataset and the evaluation metrics are provided in Appendix A.1.

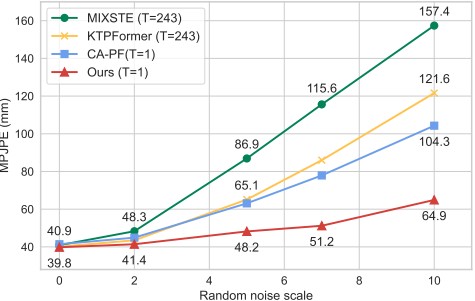

Figure 9: We add Gaussian noise with varying scales to the input 2D poses of different methods to test the robustness to noisy inputs.

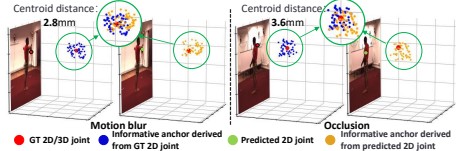

Figure 10: Adaptive 3D anchor setting comparison under inaccurate/GT 2D pose. We calculate the centroids of two anchor sets and compute the distance between them (*i.e.*, centroid distance [36]). The adaptive 3D anchors from inaccurate 2D pose closely match the GT 2D pose distribution, showing a minor centroid distance.

### 4.2 Implementation details

Our model is implemented with PyTorch. Following [55], we use pre-trained frozen HRNet-w32 [33] as image backbone and only extract the pyramid feature maps for model input. For fair and in line with previous works, we use CPN-detected [4] 2D pose as input in Human3.6M and 3DPW, GT 2D pose in MPI-INF-3DHP. The experiments are conducted on 2 NVIDIA RTX 3090 GPUs, using AdamW optimizer [24] with a learning rate of $4e^{-4}$ and weight decay of 0.98 in total 30 epochs. All ablation studies are conducted on Human3.6M [13].

### 4.3 Comparison with state-of-the-art methods

**Human3.6M dataset**. We evaluate our model with SOTA methods on Human3.6M in Table 1, covering both image based and sequence based methods. When compared to SOTA image based methods, our model demonstrates a notable improvement in MPJPE, reducing the MPJPE from 41.4*mm* to 39.8*mm* (1.6*mm* decrease), alongside a 0.8*mm* reduction in PA-MPJPE. Notably, our single-frame model matches the performance of SOTA sequence based methods, achieving an MPJPE from 40.1*mm* to 39.8*mm* without requiring temporal context. To showcase our method's superiority, we conducted a comparison with CA-PF [55] in challenging subset. Our method achieves a significant improvement by 9.3*mm* in MPJPE and 12.1*mm* in PA-MPJPE. Visual comparison under challenging subset is provided in Fig. 11. We additionally made a performance comparison for the non-challenging case in Appendix A.3.

| Method | MPJPE (Full)↓ | MPJPE (Challenging)↓ |
|---|---|---|
| Regression | 41.6 | 77.2 |
| Classification with 16 bins | 40.7 | 74.8 |
| Classification with 64 bins | 39.8 | 73.1 |
| Classification with 128 bins | 40.2 | 74.7 |

Table 6: Comparison of depth discretization strategy.

| Method | GPU Memory (M) | MPJPE ↓ |
|---|---|---|
| w/o feature sampling | 21670 | 40.0 |
| Random sampling | 13784 | 45.6 |
| 2D pose prior feature sampling | 13784 | 39.8 |

Table 7: Feature sampling strategy comparison.

**MPI-INF-3DHP dataset**. We evaluate the performance on MPI-INF-3DHP dataset in Table 2. Our PandaPose achieves the best result, outperforming the existing SOTA image based models by 0.6% in PCK, 0.4% in AUC and 0.9*mm* in MPJPE. Under challenging subset (*e.g.*row 3 in Fig. 11), our method achieved a significant advantage by 9.8% in PCK, 9.3% in AUC and 14.8*mm* in MPJPE.

**3DPW dataset**. We evaluate our model pretrained on Human3.6M to 3DPW dataset, as shown in Table 3. Under the same cross-dataset setting, our model achieves the SOTA performance with a notable improvement of 2.3*mm* in MPJPE and 1.9*mm* in PA-MPJPE, showcasing the strong generalization ability of our method.

### 4.4 Ablation study

**3D anchor setting.** To validate the effectiveness of the anchor-to-joint regression and our adaptive 3D anchor setting, we conducted ablation studies as shown in Table 4. The baseline model, which directly regresses 3D poses using an MLP from the output feature of decoder, shows a significant performance drop (MPJPE decreases by 2.3*mm* in full test set and 8.1*mm* in challenging subset compared to the anchor-to-joint regression manner). Further, both global and adaptive local anchors improve performance, with adaptive local anchors providing better accuracy due to their closer alignment with joint positions. In challenging subset, adaptive anchor improved by 2.2*mm* compared to the global anchor. When combining global and local anchors, integrating both global and local context yields the optimal performance for the model. These findings prove that an effective regression manner and carefully designed 3D anchors can enhance the performance of 3D pose estimation.

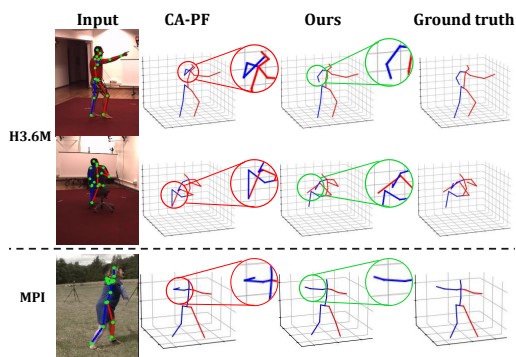

Figure 11: Visual comparison on challenging cases (*e.g.* significant occlusion or 2D pose inaccuracy). The circles highlight locations where our method has better predictions.

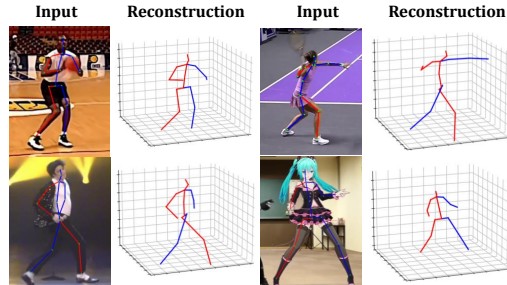

Figure 12: Visualization on samples out of dataset from Internet including various scenarios and virtual avatars.

**Joint-wise 3D feature lifting.** To assess the effectiveness of lifting 2D features to depth-aware 3D features, we conduct ablation studies on key propositions in Table 5. We first remove the entire depth branch and use only the 3D anchor and in-plane 2D feature for interaction as the baseline. Next, we predict a single depth map and add the 3D deformable cross-attention and depth cross-attention to facilitate feature lifting. The performance improved, highlighting the importance of spatial context for 3D pose estimation. Subsequently, we predict joint-wise depth distributions instead of only one single depth map. It is observed that the performance improved, especially under the challenging subset, MPJPE shows a significant improvement by 7.7*mm*, indicating that fine-grained depth information at the joint level contributes to the accuracy of 3D pose estimation, particularly under self-occlusions.

**Depth Discretization Strategy** We replace the classification depth head with a regression head for comparison, as shown in Table 6. The classification approach achieves better accuracy, with the optimal bin number being 64. The improved performance can be attributed to the better alignment between the task complexity and the lightweight classification head, which helps alleviate fitting difficulty. Especially under challenging cases with significant occlusion, the predicted MPJPE improved by 4.1*mm* (77.2 - 73.1). We also observe that using too many bins can degrade performance.

**Robustness to noisy 2D pose**. We add random Gaussian noise to the input 2D pose to compare the performance under varying noise levels in Fig. 9. As 2D pose estimate quality decreases, our method retains higher accuracy than others, demonstrating lower sensitivity to 2D pose estimator stability and greater practical flexibility. As shown in Fig. 10, our adaptive anchors demonstrate superior generalization under challenging scenarios, maintaining a distribution close to the ground truth even with inaccurate 2D poses.

**2D pose prior based feature sampling.** We verify the effectiveness of 2D pose prior based feature sampling as shown in Table 7. At the batch size 176, our strategy achieves stable performance while using nearly half the GPU memory compared to utilizing the full multi-scale feature map. In contrast, random sampling of an equal number of features leads to a significant performance drop, indicating that our approach efficiently captures essential features while reducing environmental noise.

### 4.5 Qualitative analysis

To intuitively demonstrate the superiority and generalization of our method, we evaluate our method on samples out of datasets (*i.e.*, Human3.6M and MPI-INF-3DHP) as in Fig. 12. To ensure diversity, we select different events, viewpoints, and clothing, and also test our model on virtual avatars. Our method exhibited promising results, revealing potential for real-world applications. Additional visualizations, analysis of efficiency and failure cases are provided in Appendix A.

## 5 Conclusions

In this paper, we propose PandaPose, a novel approach for 3D human pose lifting by propagating 2D pose prior to 3D anchor space as a unified intermediate representation. 3D anchor space comprises: joint-wise 3D anchors in the canonical coordinate system that offer accurate and robust priors to mitigate errors from 2D pose estimation; depth-aware joint-wise feature lifting that hierarchically integrates depth information to resolve ambiguities caused by self-occlusion; and an anchor-feature interaction decoder that combines 3D anchors with lifted features to generate unified anchor queries encapsulating joint-wise 3D anchor sets, visual cues, and geometric depth information. The anchor queries are further employed to facilitate anchor-to-joint ensemble prediction. Experiments on Human3.6M, MPI-INF-3DHP and 3DPW datasets demonstrate that PandaPose not only addresses the aforementioned challenges but also achieves state-of-the-art performance, especially under challenging scenarios. Limitations and broader impacts are discussed in Appendix A.7.

**Acknowledgment.**

This work is supported by the National Natural Science Foundation of China (Grant No. 62271221) and Taihu Lake Innovation Fund for Future Technology, Huazhong University of Science and Technology (HUST), under Grant 2023-B-8. This research is also supported by Joey Tianyi Zhou's A*STAR SERC Central Research Fund (Use-inspired Basic Research).

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

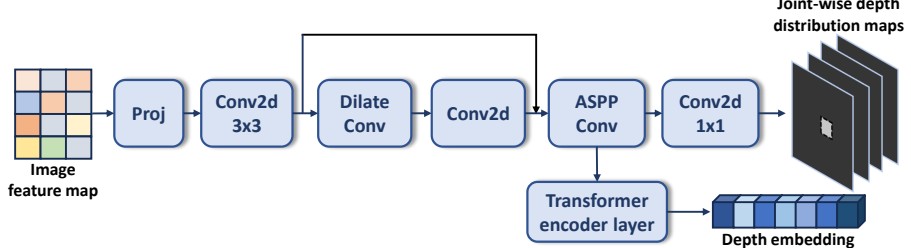

Figure 13: Illustration of light-weight depth network.

| Method | MPJPE (Challenging subset) 27377 samples | MPJPE (Non-challenging subset) 515967 samples | MPJPE (Full test set) 543344 samples |
|---|---|---|---|
| CA-PF [55] | 82.4 | 39.9 | 41.4 |
| PandaPose (ours) | 73.1 (9.3↓) | 38.8 (1.1↓) | 39.8 (1.6↓) |

Table 8: Performance comparison of different subsets on Human3.6M test set. The non-challenging subset refers to the remaining samples after excluding the challenge subset from the full test set.

# A Technical Appendices and Supplementary Material

## A.1 Datsets and evaluation metrics

**Human3.6M** [13] is a widely used benchmark for 3D human pose estimation. Following previous protocols, we train our model on 5 subjects (S1, S5, S6, S7, S8) and evaluate it on 2 subjects (S9, S11). We report MPJPE and PA-MPJPE on Human3.6M. We select samples with an error > 5 between the predicted 2D pose and GT 2D pose as the *challenging subset* (27.4k samples, about 5% in test set). These samples with 2D pose inaccuracy generally have strong occlusions or confusing pose patterns, as shown in Fig. 11.

**MPI-INF-3DHP** [26] is also widely used benchmark for 3D human pose estimation, collected in both indoor and challenging outdoor environments. We report PCK (Percentage of Correct Keypoint) with the 150 mm range, AUC (Area Under Curve) and MPJPE as evaluation metrics. Following the same principles as described above, we selected 111 samples (about 4%) as the *challenging subset*.

**3DPW** [39] is a challenging in the-wild dataset. We train our model on Human3.6M and test it on 3DPW to evaluate the generalization ability. We report MPJPE and PA-MPJPE on 3DPW.

## A.2 Depth network structure

The lightweight depth network we employ is illustrated in Fig.13. This network has approximately $5M$ parameters and is composed of multiple convolutional layers. It takes a single-layer image feature map as input, which is then processed through a feature projection layer. Subsequently, the convolutional layers extract deep features, which are further refined by an ASPP Fig.[3] (Atrous Spatial Pyramid Pooling) module to capture multi-level information. From this process, two outputs are derived:

- The depth features are passed through a 1x1 convolution to generate depth distribution maps for each joint.
- A single-layer Transformer encoder, comprised of the self-attention mechanism, processes the depth features to produce a depth embedding.

## A.3 Performance analysis on different subsets

In Table 1, we compare the performance of PandaPose with other SOTA methods on the full test set as well as on the challenging subset to demonstrate its superiority, particularly in terms of robustness to occlusions and 2D pose inaccuracies. Furthermore, it would raise a concern that improvement on difficult samples could come at the expense of weaker performance on easier cases. To address this concern, we conducted additional experiments in Table 8 by excluding the challenging subset from

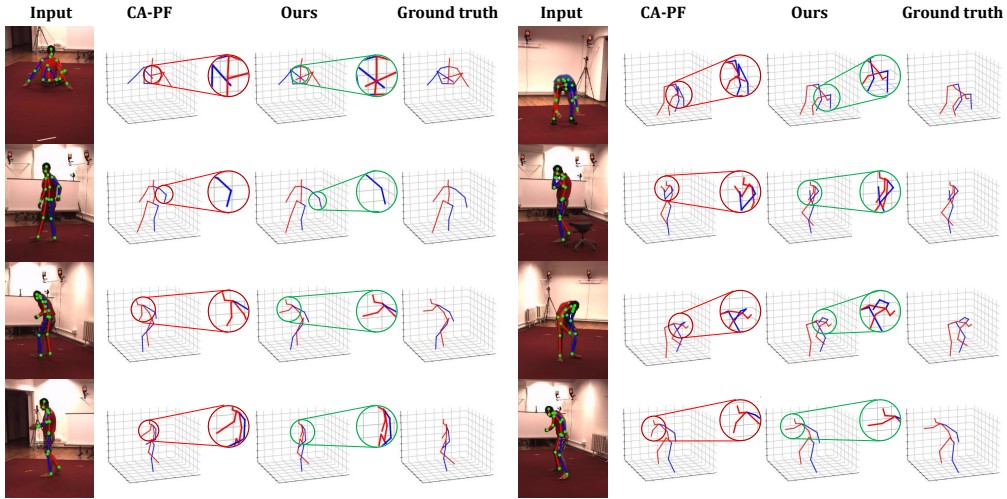

Figure 14: Visual comparison in challenging cases. The circles highlight locations where our method has better predictions.

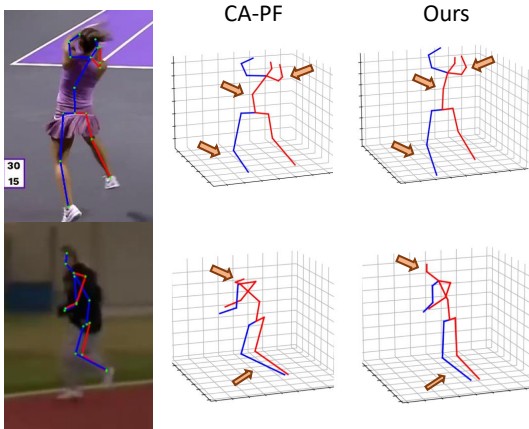

Figure 15: Failure case visualization under motion blur or self-occlusion.

the full test set, defining it as the non-challenging subset, and compared it with CA-PF [55]. Our approach exhibits notably improvements even on relatively normal cases.

### A.4 Performance analysis on different 2D pose estimator

We select three widely used 2D pose estimators (i.e., CPN [4], HRNet-w32 [33], HRNet-w48 [33]) for ablation experiments to examine how predicted 3D pose quality varies with input 2D accuracy. The results are listed in Table 10. The accuracy of PandaPose remains relatively stable with respect to the errors of the 2D pose estimator. Compared with CPN and HR-Net-w48, the 2D accuracy (mAP) decreases by 5.5%, but the error of PandaPose only increases by 1.5%. Meanwhile, in Figure 9 of the main text, we gradually add Gaussian noise to the 2D pose to study the method tolerance to noise levels and our method maintains the best anti-noise ability. When the pixel-level noise scale ranges from 0 to 5, the error of PandaPose only increases by 16% (from 39.8 to 48.2), while the image-based SOTA CA-PF [55]increases by 56% (from 41.4 to 64.9), and the sequence-based SOTA KTPFormer [28] increases by 112% (from 40.1 to 86.9). The noise resistance can be attributed to different 3D anchor setting and anchor-to-joint ensemble prediction mechanism.

### A.5 Running efficiency comparison

Table 9 presents an efficiency comparison among our proposed method (PandaPose), the state-of-the-art image based method (CA-PF [55]) and sequence based methods (MixSTE [50] and MHFormer [20]) on Human3.6M [13]. All metrics are evaluated on an NVIDIA RTX3090 GPU.

| Method | FPS | Lifting Module Parameters($\mathbf{M}$) | MPJPE ↓ (Full) | MPJPE ↓ (Challenging) |
|---|---|---|---|---|
| Sequence based | | | | |
| MHFormer(T=27) [20] | - | 24.8 | 45.9 | - |
| MHFormer(T=351) [20] | - | 24.8 | 43.0 | - |
| MixSTE(T=243) [50] | - | 33.6 | 40.9 | - |
| Image based | | | | |
| CA-PF [55] | 19 | 14.1 | 41.4 | 82.4 |
| PandaPose w/ 3 decoder | 16 | 15.2 | **39.8** | **73.1** |
| PandaPose w/ 2 decoder | 18 | 12.7 | **40.3** | **75.6** |

Table 9: Running efficiency comparison.

| 2D pose estimator | mAP on COCO ↑ | MPJPE on Human 3.6M ↓ |
|---|---|---|
| CPN | 68.6 | 40.1 |
| HR-Net-w32 | 74.4 | 39.8 |
| HR-Net-w48 | 76.3 | 39.5 |

Table 10: Comparison of 2D pose estimators on COCO and Human 3.6M datasets.

Due to the differences in experimental setup, we omit the FPS metric for the sequence-based methods. Compared to the sequence-based methods, PandaPose takes fewer parameters while maintaining comparable performance. To demonstrate the superiority of PandaPose, we reduce the number of layers in the decoder from 3 to 2. This adjustment result in a parameter reduction of $2.5M$, with only a performance drop of $0.5mm$ on the full test set and $2.5mm$ on the challenging subset. Compared to CA-PF [55], PandaPose reduces parameters by $1.4M$ while maintaining comparable running efficiency. Notably, on the challenging subset, PandaPose outperforms CA-PF by $6.8mm$ MPJPE. This also indicates that PandaPose holds essential potential to further efficiency enhancement, but still ensuring promising performance.

## A.6 More visualization results

We present more visualization cases in Fig. 14 and compare them with state-of-the-art image based method (CA-PF [55]) and ground truth. Our method demonstrates superior handling of the relative depth relationships between joints when dealing with severe self-occlusion or noisy 2D pose inputs, resulting in more accurate 3D pose predictions.

Additionally, some representative failure cases are shown in Fig. 15. In scenarios with severe motion blur or self-occlusion, the human subject may become confused with the background, leading to inaccurate predictions of 2D poses. Thus, the quality of 3D pose predictions will be affected. Our method, due to its explicit modeling of the relative depth relationships between joints and the integration of anchor-to-joint ensemble prediction, is capable of predicting relatively more reasonable 3D human poses in such scenarios.

## A.7 Limitations and boarder impacts

Despite the achievements of our method in image based 3D human pose estimation, several limitations remain that are worth further improvement. As a single-frame method, PandaPose lacks temporal smoothness compared to sequence-based methods, which use information from adjacent frames. This can lead to jitter in pose estimation during continuous actions, especially with rapid or complex movements. Additionally, PandaPose introduces the processing of image features and complex operations including feature lifting and ensemble prediction in the 3D anchor space. While these steps improve accuracy, they inevitably come with a computational resource cost. Future research will aim to develop more lightweight approaches with similar performance but reduced computational and memory costs, exploring efficient feature lifting and optimization methods. Our approach exclusively utilizes publicly available datasets during the training process, thereby having no broad societal impact, not involving AI ethics, and not involving any privacy-sensitive data.

