# OpenReview forum: "PandaPose: 3D Human Pose Lifting from a Single Image via Propagating 2D Pose Prior to 3D Anchor Space"
_NeurIPS.cc/2025/Conference — NeurIPS 2025 poster_

### Official Review · Reviewer_Cg4S · 2025-06-02

**Clarity:** 3
**Significance:** 3
**Originality:** 3
**Rating:** 4
**Confidence:** 4

**Summary:**

The authors propose a method to lift 2D pose estimations to 3D. This method attempts to mitigate the problem of self-occlusion and depth ambiguity by careful model design. The experimental verification demonstrates the effectiveness of the proposed methods.

**Questions:**

Please address the questions in Weaknesses.

**Ethical Concerns:**

["NO or VERY MINOR ethics concerns only"]

**Final Justification:**

The authors' rebuttal addresses my concerns, and considering the contribution and strength of this paper, I keep my rating.

**Limitations:**

Yes

**Quality:**

3

**Strengths And Weaknesses:**

The authors design a new method for 2D-to-3D pose lifting. They propose several interesting components:
- an extended set of joint anchors
- separate joint depth distribution modeling
- a feature fusion strategy that leverage 2D joint and depth estimation.

Through the experiments on public dataset, they demonstrate the effectiveness of this method with the lowest reconstruction errors. The ablation study is thorough and show the necessity of each design. The results on 3DPW dataset also demonstrate the robustness and the generalization ability of the proposed method.

Generally, the paper is detailed and well structured.

For the Weaknesses, there are several points needed to be addressed:
- In Eq. 1, what is `Linear`? Is it a neural network? If so, is it optimized in the training of the whole pipeline?
- For the plane of root joint in Line 151, is it fixed to `z=0`?
- In the training, are the GT 2D points or the estimated 2D points used?
- In Eq. 6, what are these sampled points? How are they generated? And how to estimate the offset?
- What is the noise level in Table 9? The unit of noise scale is not clear.
- Providing some qualitative results on 3DPW would benefit the evaluation, as it has GT and is in less constrained outdoor setting.
- For other methods shown in Table 3, are they trained on 3DPW?
- Since the experiments are conducted with the 2D pose estimator [33], I'm wondering whether the results will change a lot if using different pose estimators?

---

> ### Author Rebuttal · Authors · 2025-07-31
>
> We sincerely appreciate your positive feedback regarding the novelty and effectiveness of our method and your recognition of the detailed and well-structured writing. In the following, we address your questions point by point.
>
> > - In Eq. 1, what is `Linear`? Is it a neural network? If so, is it optimized in the training of the whole pipeline?
> >
>
> `Linear` means a linear projection layer responsible for estimating 2D joint to 3D anchor offsets $\\delta_J$from the input 2D pose $P\_J^{2D}$, and is optimized end-to-end during the training process.
>
> > - For the plane of root joint in Line 151, is it fixed to `z=0`?
> >
>
> Yes, follow the common practice of 3D human pose estimation methods (CA-PF\[NeruIPS'23\], HiPART\[CVPR'25\], KTPFormer\[CVPR'24\] etc.), we set the root joint depth $z$ to 0.
> We will add the explanation regarding this in Line 151.
>
>
> > - In the training, are the GT 2D points or the estimated 2D points used?
> >
>
> We strictly follow the same input as the previous methods (CA-PF\[NeruIPS'23\], HiPART\[CVPR'25\], KTPFormer\[CVPR'24\] etc.) for fair comparison, i.e., using the same estimated 2D pose for training and testing (Line 269-272).
>
>
> > - In Eq. 6, what are these sampled points? How are they generated? And how to estimate the offset?
> >
>
> $$DCA(a) = \\sum\_{n \\in N} W\_n \\phi(F\_{3D}, P\_a + \\Delta S\_n)$$
> For an input anchor query $Q\_{anchor}$ and corresponding anchor position $P\_a$, we treat $P\_a$ as reference point to execute deformable attention\[1\]. Aligned with \[1\], we equip each anchor with $N$ sampling points nearby to dynamically interact with $F\_{3D}$. These sampled points are generated by adding *sampling offsets* $\\Delta S\_n$to the reference points $P\_a$ (i.e. 3D anchor positions). The sampling offsets $\\Delta S\_n$ are predicted from the input queries (also referred as anchor queries $Q\_{anchor}$ in our paper) through a linear projection layer. This design enables the Transformer to dynamically focus on sparse but semantically meaningful local regions in an adaptive manner, improving both efficiency and accuracy.
> Due to space limitations, we will add detailed explanations for Eq.6 in the revised version, and add a flowchart for the signal to enhance clarity.
>
>
> > \[1\] Deformable DETR: Deformable Transformers for End-to-End Object Detection
>
>
> > - What is the noise level in Table 9? The unit of noise scale is not clear.
> >
>
> The noise level in Table 9 is measured in pixel units, based on images of resolution 256×192.
>
>
> > - Providing some qualitative results on 3DPW would benefit the evaluation, as it has GT and is in less constrained outdoor setting.
> >
>
> We agree that qualitative results on 3DPW would be valuable for demonstrating the effectiveness and generalization of our method, especially in unconstrained outdoor settings. Unfortunately, due to the constraints of the rebuttal policy, we are unable to include additional visualizations at this stage. We will include more qualitative results on 3DPW in the revised version to further support the evaluation.
> In the meantime, to demonstrate the generalization ability of outdoor scenes, we present the comparison results of MPI-INF-3DHP with SOTA and GT (Figure 11) as well as the qualitative results of in-the-wild internet images (Figure 12).
>
> > - For other methods shown in Table 3, are they trained on 3DPW?
> >
>
> The methods listed in Table 3 are all 3D human pose estimation approaches that follow the cross-dataset setting (training on Human3.6M and testing on 3DPW). The performance metrics of these methods are taken from their latest publicly available reports. We compare against them under the same evaluation protocol to ensure a fair comparison. Compared to the latest SOTA method HiPART (CVPR 2025), we achieve significant improvements in performance (74.9 vs 77.2, 2.3 $\\downarrow$), demonstrating the strong generalization capability of our approach.
>
> > - Since the experiments are conducted with the 2D pose estimator \[33\], I'm wondering whether the results will change a lot if using different pose estimators？
> >
>
> We select three widely used 2D pose estimators (i.e., CPN, HRNet-w32, HRNet-w48) for ablation experiments to examine how predicted 3D pose quality varies with input 2D accuracy. The results are listed below
>
> | **2D pose estimator** | **mAP on COCO $\uparrow$** | **MPJPE on Human 3.6M $\uparrow$** |
> | --- | --- | --- |
> | CPN | 68.6 | 40.1 |
> | HR-Net-w32 | 74.4 | 39.8 |
> | HR-Net-w48 | 76.3 | 39.5 |
>
> The accuracy of PandaPose remains relatively stable with respect to the errors of the 2D pose estimator. Compared with CPN and HR-Net-w48, the 2D accuracy (mAP) decrease by 5.5%, but the error of PandaPose only increase by **1.5%.**
> Meanwhile, in Figure 9 of the main text, we gradually add Gaussian noise to the 2D pose to study the method tolerance to noise levels and our method maintains the best anti-noise ability. When the pixel-level noise scale ranges from 0 to 5, the error of PandaPose only increases by **16%** (from 39.8 to 48.2), while the image-based CA-PF increases by **56%** (from 41.4 to 64.9), and the sequence-based SOTA KTPFormer increases by **112%** (from 40.1 to 86.9).
> The noise resistance can be attribute to different 3d anchor setting and anchor-to-joint ensemble prediction mechanism.

---

> > ### Comment · Reviewer_Cg4S · 2025-08-05
> >
> > Thanks for the detailed rebuttal to address my concerns. Please revise the paper accordingly to include them.

---

> > > ### Author Response · Authors · 2025-08-05
> > >
> > > We express our gratitude to the reviewer for the feedback! We will carefully incorporate your suggestions to improve our paper.

---

### Official Review · Reviewer_HbVK · 2025-06-23

**Clarity:** 3
**Significance:** 3
**Originality:** 3
**Rating:** 4
**Confidence:** 3

**Summary:**

This paper introduces PandaPose, a novel approach for 3D human pose estimation from a single RGB image. The key innovation lies in propagating 2D pose priors into a 3D anchor space, which consists of joint-wise 3D anchors, depth-aware feature lifting, and an anchor-feature interaction decoder. The method addresses two major challenges in existing approaches: error propagation from noisy 2D inputs and depth ambiguity in self-occluded cases. Experiments on Human3.6M, MPI-INF-3DHP, and 3DPW demonstrate state-of-the-art performance, with significant improvements in challenging scenarios.

**Questions:**

1. How was the number of 3D anchors determined? Do the learned anchors exhibit specific semantic properties, such as spatial discrimination or differentiation between certain types of 3D points?

2. Possible typo: Should "Figure 13" be "Table 6"? While the 2D feature sampling strategy reduces GPU memory and computational costs, it does not lead to significant performance improvement. This raises a question: If the 2D pose estimation contains large errors, could the sampled features deviate from the desired content? If so, why does the performance not degrade noticeably?

**Ethical Concerns:**

["NO or VERY MINOR ethics concerns only"]

**Final Justification:**

Thank you for the author's response. I will keep my score.

**Limitations:**

Yes.

**Quality:**

3

**Strengths And Weaknesses:**

**Strengths:**
1. The paper is clearly written, with a logical structure and well-organized presentation.
2. Well-Motivated Design. The use of joint-wise 3D anchors and depth-aware lifting addresses known weaknesses in previous pose lifting works that rely on fixed mappings or 2D-plane representations.
3. Strong Empirical Results. The method achieves SOTA performance across multiple benchmarks, with notable improvements. The ablation studies convincingly validate the contributions of each component.

**Major Weaknesses:**
1. Depth Discretization Strategy. Although using classification for depth estimation is pragmatic, it might introduce quantization errors. Could a regression-based approach be adopted for this task? If feasible, it would be valuable to include a performance comparison between the two approaches.
2. Computational Cost. The paper introduces additional intermediate representations and extra computational steps for image-keypoint depth feature interactions. What is the inference speed of the proposed algorithm? Does it meet real-time requirements?

---

> ### Author Rebuttal · Authors · 2025-07-31
>
> We appreciate the recognition by Reviewer HbVK of our writing, well-motivated design, and empirical results. We will address all concerns regarding weaknesses and questions in the following.
>
> ## **Depth Discretization Strategy**
>
> Thanks for the insightful suggestion. We simply replace the classification depth head with a regression head for comparison as shown in table below. The classification approach achieves better accuracy, with the optimal bin number being 64. The improved performance can be attributed to the better alignment between the task complexity and the lightweight classification head, which helps alleviate fitting difficulty — a benefit particularly evident in challenging cases. We also observe that using too many bins can degrade performance, which aligns with findings in previous lightweight depth estimation methods \[1,2,3\]. We will include these results and discussions in the revised version to further clarify the advantages of our method.
> We kindly note that our insight regarding the depth head does not lie in whether it is for classification or regression, but rather in the joint-wise approach. Ordinary predicting single depth map in principle cannot reflect joint depth in cases of self-occlusion (as shown in Figure 5 of the main text). The proposed joint-wise module addresses this issue by independently predicting the depth distribution for each joint, improving robustness as shown in Table 5 row2 - row3 of main text. Especially under challenging cases with significant occlusion, the predicted MPJPE decreased by 2.8mm (75.9 - 73.1).
>
> |  | MPJPE on Human3.6M| Challenge case |
> | --- | --- | --- |
> | Regression | 41.6 | 77.2 |
> | Classification with 16 bins | 40.7 | 74.8 |
> | Classification with 64 bins | 39.8 | 73.1 |
> | Classification with 128 bins | 40.2 | 74.7 |
>
>
>
> > \[1\] Deep ordinal regression network for monocular depth estimation, CVPR2018
> >
> > \[2\] Adabins: Depth estimation using adaptive bins, CVPR2021
> >
> > \[3\] idisc: Internal discretization for monocular depth estimation， CVPR2023
>
>
> ## **Computational cost**
>
> In Table 7 of the appendix, we compare the parameters and FPS with the SOTA image-based method CA-PF, as shown in the following table (assuming real-time application requires the single-frame input). Our method demonstrates significantly superior accuracy, especially in challenging cases, while FPS and inference consumption remain at a similar level to CA-PF. Following their claim of efficiency, PandaPose also meets the real-time requirement.
>
> | method | FPS | Param | Inference GPU memory | MPJPE on Human3.6M (full) | MPJPE on Human3.6M (challenging)|
> | --- | --- | --- | --- | --- | --- |
> | CA-PF (T=1) | 19 | 14.1 | 310 | 41.4 | 82.4 |
> | PandaPose w/3 decoder (T=1) | 16 | 15.2 | 323 | 39.8 | 73.1 |
> | PandaPose w/2 decoder (T=1) | 18 | 12.7 | 307 | 40.3 | 75.6 |
>
>
> ## **Number setting and semantic characteristics of 3D anchor**
>
>
> Regarding the **the number setting of 3D anchors**, we determine it based on the trade-off between performance and computational overhead. We conduct ablation experiments under different anchor numbers (see the table below) and find that increasing the anchor number from 149 to 596 continuously improve performance; however, when it is further increased beyond 596, the performance improvement become limited, while the computational overhead increase significantly. Therefore, we finally set the anchor number to 596 as the balance point between performance and efficiency.
>
> | Anchor numbers | MPJPE on Human3.6M | GFlops |
> | --- | --- | --- |
> | 149 | 40.5 | 14.2 |
> | 596 | 39.8 | 16.6 |
> | 1022 | 39.7 | 18.9 |
>
> Regarding the **semantic characteristics of anchor**, although we are unable to present the visualization results of anchor in the paper due to the rebuttal policy, we have verified through quantitative and qualitative analysis whether its spatial distribution has semantic significance.
> Specifically, we qualitatively and quantitatively observe the distribution and contribution of different types of anchors (adaptive local anchor / fixed global anchor) to the final joint prediction (as shown in Figure 3 of the main text). By selecting top-50 anchor-to-joint weights, we identify informative anchors contributing most to joint prediction. (Line 139-141) The distribution of the final informative anchors shows obvious clustering for the target joint (mean offset 73.4mm), with offsets to other joints all significantly larger than this value (mean offset 161.9 mm), demonstrating the specificity of the distribution.
> We also analyze the changes in the offset of each anchor to the corresponding joint during the training process. This offset shows a clear downward trend during training as shown in the table below, indicating that the model gradually learns the accurate correspondence between anchors and 3D joints. This further demonstrates that anchors are not randomly distributed but gradually converge to positions consistent with the semantics of the human body structure.
>
> | Training step | Mean 3D anchor-to-joint offset |
> | --- | --- |
> | 10 | 259.12 mm |
> | 100 | 101.9 mm |
> | 1000 | 72.8mm |
>
> In summary, our analysis shows that the learned 3D anchors are not only effective in terms of performance but also possess certain semantic significance. We will further supplement these analyses in the revised manuscript to more clearly demonstrate the design motivation and characteristics of the anchors.
>
>
> ## **2D feature sampling stratery**
>
> Thanks for the insightful comment. We have considered this issue and implemented targeted optimizations in the design of **feature sampling strategy** and **anchor mechanism**:
>
> 1. For **feature sampling strategy**, based on the four-level multi-scale feature maps after downsampling, we perform feature sampling only on the lower two levels while preserving global semantic feature from two lower resolution levels (Line 205-207). Since 2D pose coordinates are inherently less sensitive to prediction errors after downsampling, we also enhance robustness to localization inaccuracies by sampling features within an r×r neighborhood around each 2D joint (Line 202-203).
>
> 	  To further demonstrate the optimizations above, we conducte an investigation into the similarity of visual features and the impact on final performance under noisy 2D pose inputs as table below. The resolution of the image is 256\*192. When the noise scale is controlled within a certain range (0-5 pixels), the sampled image feature similarity can maintain relatively high stability (100% - 95.21%). When noise overflows to a large extent (10 pixels), the image features will deviate to a certain degree (85.33%) but still at a relatively high similarity.
>
> 2. Apart from the optimization of the feature sampling strategy, we additionally adopt the **anchor mechanism** based on local and global anchors to ensure that our model can provide reliable 3D predictions even when there are significant errors in 2D pose estimation.
> 	*  For Local Anchors, we sample a set of 3D local anchors for each 2D joint using learnable sampling offsets. As shown in Figure 10 of the main text, even when 2D pose estimations exhibit substantial deviations due to motion blur or severe self-occlusion, these local anchors still effectively capture the positions of 3D joints.
>
> 	* For Global Anchors, we established a series of fixed global anchors that cover the potential human pose space (Line 149-152). This ensures that, despite significant shifts in local feature details caused by pose estimation errors, the global anchors can serve as stable reference points to provide global context, thereby maintaining the quality of the final 3D pose estimation.
>
> 	* Additionally, our method does not solely rely on a single offset prediction, but instead adopts an ensemble approach for offset prediction. This means that for each anchor, we generate multiple offset predictions and determine the final position through weighted averaging (Eq.7). This method not only improves the robustness of the prediction but also effectively reduces the overall performance degradation caused by inaccurate individual predictions.
>
>
> To demonstrate the effectiveness and superiority of the above optimizations. We compare with other SOTA methods to test the robustness to noisy inputs (see table below and Figure 9 in main text). When the noise scale is 10 pixels, our MPJPE will decrease by 63.8%, but the magnitude is still not as severe as that of CA-PF by 151.9% and MixSTE by 284.8%.
>
> | Noise scale (pixel-level）| Sampled feature cosine similarity | MPJPE（Ours） | MPJPE (CA-PF) | MPJPE (MixSTE)|
> | --- | --- | --- | --- | --- |
> | 0 | 100% | 39.8 | 41.4 | 40.9 |
> | 2 | 98.61% | 41.4 | 43.5 | 48.3 |
> | 5 | 95.21% | 48.2 | 64.9 | 86.9 |
> | 7 | 91.37% | 51.2 | 78.3 | 115.6 |
> | 10 | 85.33% | 64.9 | 104.3 | 157.4 |
>
> Finally, we would like to thank you for the typo in Fig13, and we will update this to Table 6 in the revised version.

---

### Official Review · Reviewer_G3sQ · 2025-06-24

**Clarity:** 4
**Significance:** 3
**Originality:** 3
**Rating:** 4
**Confidence:** 5

**Summary:**

PandaPose presents a new approach for 3D human pose estimation from a single RGB image, tackling issues like errors from 2D pose predictions and self-occlusion. Key innovations include setting individual 3D anchors for each joint, depth-aware feature lifting to fix depth issues, and a decoder that combines different features for better joint prediction. It also uses joint-specific depth distributions for more accurate depth modeling and a smart feature sampling method based on 2D pose priors. Tests show it outperforms existing methods, especially in complex scenarios like occlusion.

**Questions:**

As shown in the Weaknesses section, A reasonable explanation will make me raise the score for this work.

**Ethical Concerns:**

["NO or VERY MINOR ethics concerns only"]

**Final Justification:**

My concerns have been resolved.

**Limitations:**

Yes

**Quality:**

3

**Strengths And Weaknesses:**

# Strengths
A series of  modules based on anchor points, which work in coordination to achieve excellent performance in 3D human pose estimation:
- Joint-wise 3D Anchors: Combine the adaptive anchors generated by the input 2D poses through a network with the fixed anchors.
- Joint-wise Depth Estimation: Instead of predicting a single depth value, PandaPose estimates depth distributions for each joint, addressing the common issue of depth ambiguity in closely placed joints, which makes it more accurate in challenging scenarios.
- Efficient Feature Sampling: The 2D pose prior-based feature sampling reduces computation by focusing on relevant areas around joints, which not only makes the process more efficient but also helps improve the overall accuracy by filtering out unnecessary background noise.
- Anchor-Feature Interaction Decoder: The decoder uses a multi-modal attention mechanism, combining visual (from Efficient Feature Sampling), depth (from Joint-wise Depth Estimation), and geometric (from Joint-wise 3D Anchors) features. This helps PandaPose enhance joint localization, particularly when parts of the body are occluded.
This anchor-based system fully exploits the 2D pyramid features and achieves better performance.

# Weakness
## Major
- This anchor-based system appears logically sound. However, it remains unclear what advantages anchors offer compared to directly predicting the offset of the 3D root node, and why such advantages exist. The author should provide a comprehensive analysis or reasoning for this design choice around line 310. This is my primary concern.
## Minor
- What's the initial Qanchor in line 218? Is it the embedding from the anchor offset? That's not clear.

---

> ### Author Rebuttal · Authors · 2025-07-31
>
> We would like to thank Reviewer G3sQ for the recognition of innovations, writing clarity and excellent performance. We also sincerely appreciate your attention to the anchor-based design in our method.
> We will comprehensively address your concerns from multiple perspectives: paradigm comparison, principle analysis spanning distribution, training stability, and ensemble mechanism, as well as relevant quantitative results.
>
> 1. **Inclusiveness of the paradigm**: `directly predicting the offset of the 3D root node` can be seen as a special case of our method. Anchor-based PandaPose initially sets global fixed anchor and the joint-wise local anchor, then combines the two types of anchors to regress the offset for obtaining 3D joint predictions. When the joint-wise anchor is discarded and the global fixed anchor is set to the root node with its number decrease to 1, PandaPose degenerates into the method you describe. We quantitatively evaluated this strategy, yielding an MPJPE of 42.7. Our final experimental results showed a 6.8% reduction in error compared to this baseline.
>
>
> | **Anchor strategy** | **MPJPE on Human3.6M** |
> | --- | --- |
> | Directly predicting the offset of the 3D root node | 42.7 |
> | Global anchor | 40.8 |
> | Ours (Global anchor + Local anchor) | 39.8 |
>
> In the revised version, we will supplement this comparison to Table 4 in the main text to further improve the completeness of the experiment.
>
> 2. **Analysis of the essential principle**:
> 	* The global fixed anchor and joint-wise anchor we propose efficiently deliver a robust initialization for 3D pose lifting that is closer to the target joint distribution compared to the root node. Subsequently, the final prediction can be obtained by simply predicting an offset with a smaller numerical range ( 91.3mm v.s. 456.2mm in world coordinate system, based on average offset statistics). This substantially lowers the difficulty of directly predicting from the 3D root node to the target 3D joint.
>
> 	* On the other hand, a smaller predicted offset also makes network training more stable and the convergence speed faster. Specifically, under the exact same dataset (Human3.6M) and computing environment, aligning the training settings, when the training MPJPE loss converges to 60mm, our method requires about 280 iterations, the root node strategy requires about 510 iterations, and CA-PF requires about 1350 iterations.
>
> 	* Additionally, our anchor mechanism essentially serves as an ensemble prediction for the joint, generating the coordinates of the target joint by integrating the offset/weight of multiple anchors.  This characteristic of "ensemble" makes the overall prediction more fault-tolerant to local noise, which has also been widely proven effective for performance improvement [1] and robustness against outliers [2].
>
> | Method | Ours | Root node strategy | CA-PF |
> | --- | --- | --- | --- |
> | Iterations when training MPJPE converges to 60mm | 280 | 510 | 1350 |
>
> We will also supplement the paper with the detailed discussion above to more comprehensively explain the advantages of our motivation and anchor design.
>
> 3. **Justification of** $Q\_{anchor}$**.** The initial $Q\_{anchor}$ is derived from the 3D anchor set. We map it to a high-dimensional space through a linear layer, thus encoding it as a **learnable anchor query,**  which is used to interact with RGB & depth features in the decoder. This is also mentioned in Line 221-224 of the main text.
>
>
> > \[1\] ediff-i: Text-to-image diffusion models with an ensemble of expert denoisers
> >
> > \[2\] Uncertainty-based offline reinforcement learning with diversified q-ensemble

---

> > ### Comment · Reviewer_G3sQ · 2025-08-03
> >
> > # Response
> > Q1: Accept
> > Q2: To ensure my understanding is accurate, does the value of Qanchor come from the 3D absolute coordinates of the anchor?
> >
> > # About comments from other reviewers
> > Other reviewers have deeper insights. Referring to the author's response, I personally believe that most of the issues have been resolved, except for: (Reviewer 2274) More ablation study about 3D anchor interaction decoder, i.e., the necessity of the three attentions.

---

> > > ### Author Response · Authors · 2025-08-05
> > >
> > > Thanks for your active response. We are glad that our analysis of the 3D anchor design addresses your concern (Q1). Below, we respond to your questions regarding $Q\_{anchor}$ and the decoder module ablation.
> > >
> > > ## Details about $Q\_{anchor}$
> > >
> > > The $Q\_{anchor}$ values are derived from the 3D absolute coordinates of the anchors, specifically by mapping from (num\_anchor, 3) to (num\_anchor, hidden\_dim) through a linear layer. We will further expand the explanation in Line 221 accordingly.
> > >
> > > ## Ablation study on the decoder
> > >
> > > In our initial response to Reviewer 2274, we focus on justifying the enhancements we introduce to the standard deformable transformer decoder ( i.e., the depth cross-attention and 3D deformable attention)—designed for depth-aware modeling. To more clearly and thoroughly illustrate the individual contributions of the three attention modules, we further conduct ablation studies starting from the simplest baseline. The expanded results on Human3.6M are provided in the table below.
> > >
> > > | Row index | Depth cross-attention | Anchor self-attention | Deformable cross-attention | MPJPE (full) | MPJPE (challenging) |
> > > | --- | --- | --- | --- | --- | --- |
> > > | 1 | \- | \- | \- | 44.9 | 86.8 |
> > > | 2 | \- | $\\checkmark$ | \- | 42.7 | 84.7 |
> > > | 3 | \- | $\\checkmark$ | 2D | 40.9 | 80.8 |
> > > | 4 | $\\checkmark$ | \- | 2D | 41.2 | 80.3 |
> > > | 5 | $\\checkmark$ | $\\checkmark$ | 2D | 40.7 | 79.5 |
> > > | 6 | $\\checkmark$ | \- | 3D | 40.4 | 76.0 |
> > > | 7 | \- | $\\checkmark$ | 3D | 40.3 | 76.4 |
> > > | 8 | $\\checkmark$ | $\\checkmark$ | 3D | **39.8** | **73.1** |
> > >
> > > The method in Row 1 refers to directly predicting the anchor-to-joint offset/weight from the input $Q\_{anchor}$ . It can be seen that all three attention modules directly improve prediction performance. Here we highlight some impressive points:
> > >
> > > - From row 2 - row 3, the 2D deformable cross-attention allows anchor features to interact with visual features, which significantly enhances the semantic nature of anchors and prediction performance (42.7 - 40.9).
> > >
> > > - From row 6 - row 8, the introduction of anchor self-attention helps to establish the connection in anchor self articulation, while also exerting a positive impact on performance (40.4 - 39.8).
> > >
> > > - From row 7- row 8, depth cross-attention explicitly equips anchor features with depth domain characteristics, resulting in performance improvement (40.3 - 39.8). Notably, particularly in challenging cases, notable enhancements are observed (76.4 - 73.1).
> > >
> > > - From row 5 - row 8, the 2D deformable attention is replaced with 3D deformable attention. The performance is further improved (40.7 - 39.8) due to the direct addition of depth distribution for depth-visual-anchor feature interaction. Especially in challenging cases, the introduction of depth features is very helpful in alleviating depth ambiguity and self-occlusion (79.5 - 73.1).
> > >
> > >
> > > These also align with the conclusions presented in Table 5, Line 311-319 of the main text.
> > >
> > > We highly appreciate your professional comments, recognition of our response, and responsibly referring to the opinions of other reviewers to summarize, supplement, and participate in discussions. We sincerely hope that our response can directly address the remaining concerns. If there are any concerns or questions, we are glad to raise further discussion!

---

### Official Review · Reviewer_MoUK · 2025-07-01

**Clarity:** 3
**Significance:** 3
**Originality:** 4
**Rating:** 4
**Confidence:** 4

**Summary:**

This work introduces PandaPose, a novel monocular 3D human pose estimation framework. In its first stage, PandaPose employs a pre-trained 2D pose detector to extract 2D joint coordinates from a single RGB image. The methodological innovation resides entirely in the second stage, which focuses on precisely and robustly estimating 3D poses by leveraging both the 2D coordinates and their associated intermediate visual features. Departing from conventional approaches that directly lift 2D poses to 3D representations, PandaPose designs a sophisticated lifting network that: (a) incorporates intermediate features from the 2D network alongside joint coordinates, and (b) introduces a 3D anchor space to jointly resolve depth ambiguity.

**Questions:**

As noted above, could the following be addressed in the revision:
1. Provide detailed comparative data on computational and memory cost.
2. Investigate the framework’s tolerance to 2D pose estimation inaccuracies—specifically, conduct ablation studies to analyze how different 2D detectors impact final 3D accuracy? This analysis should verify whether the system maintains consistent 3D pose outputs when input 2D precision varies marginally.

**Ethical Concerns:**

["NO or VERY MINOR ethics concerns only"]

**Final Justification:**

I've carefully reviewed the comments of the other reviewers. I believe the weaknesses raised by reviewer 2274 are not that critical. The rebuttal solved most of my concerns, and I'll keep my rating.

**Limitations:**

Yes

**Quality:**

3

**Strengths And Weaknesses:**

Strengths:
1. ​​Innovative Methodology​​: The framework integrates (a) an adaptive 3D anchor space and (b) a multimodal feature interaction decoder, demonstrating notable methodological novelty.
​​2. Superior Performance​​: Achieves state-of-the-art (SOTA) results on standard benchmarks, with efficacy further validated through a constructed "challenging subset".

Weakness:
​​1. Efficiency Limitations​​: While acknowledging extra computational and memory overhead in the appendix, parameter count alone fails to adequately characterize efficiency. We recommend supplementing with concrete quantitative metrics (e.g., GFLOPs) visualized via scatter plots to explicitly illuminate the accuracy-efficiency trade-offs.

---

> ### Author Rebuttal · Authors · 2025-07-31
>
> We appreciate Reviewer MoUK recognizing the novelty and superior performance in our method. Below, we address your concerns from two perspectives:
>
> ## **Efficiency analysis**
>
> | method | FPS | Param | Inference GPU memory cost（M） | GFLOPs | MPJPE on Human3.6M (full) | MPJPE on Human3.6M (challenging) |
> | --- | --- | --- | --- | --- | --- | --- |
> | MixSTE (T=81) | \- | 33.6 | 192 | 46.3 | 42.4 | \- |
> | MixSTE (T=243) | \- | 33.6 | 295 | 139.1 | 40.9 | \- |
> | CA-PF (T=1) | 19 | 14.1 | 310 | 8.2 | 41.4 | 82.4 |
> | PandaPose w/3 decoder (T=1) | 16 | 15.2 | 323 | 16.6 | 39.8 | 73.1 |
> | PandaPose w/2 decoder (T=1) | 18 | 12.7 | 307 | 15.4 | 40.3 | 75.6 |
>
> We supplement the GFLOPS metric and inference memory cost based on Table 7 in the appendix. Our method achieves superior performance with significantly fewer FLOPs compared to sequence-based MixSTE. Compared to the previous image-based SOTA CA-PF, although our method has slightly higher GFLOPs (still at an absolutely low level compared with MixSTE), it benefits from the highly parallel design of the attention implementation, resulting in comparable FPS and GPU memory cost. Moreover, our method demonstrates advantages in overall accuracy, especially in challenging cases, indicating its high efficiency and effectiveness.
> Due to the limitations of the rebuttal policy, we can only provide tables instead of scatter plots, and we will add scatter plots corresponding to the above data in the revised version. Overall, PandaPose achieves the best accuracy-efficiency balance.
>
>
> ## **Tolerance to 2D pose estimation inaccuracies**
>
> We also thank the reviewer for the advice on evaluation of the tolerance to 2D pose estimation inaccuracies.
> We select three widely used 2D pose estimators (i.e., CPN, HRNet-w32, HRNet-w48) for ablation experiments to examine how predicted 3D pose quality varies with input 2D accuracy. The results are listed below
>
> | **2D pose estimator** | **mAP on COCO(**$\\uparrow$**)** | Method | **MPJPE on Human 3.6M(full)** | **MPJPE on Human 3.6M(challenging)** |
> | --- | --- | --- | --- | --- |
> | CPN | 68.6 | CA-PF | 41.6 | 83.7 |
> | CPN | 68.6 |  Ours | **40.1** | **73.7** |
> | HR-Net-w32 | 74.4 | CA-PF | 41.4 | 82.4 |
> | HR-Net-w32 | 74.4 | Ours | **39.8** | **73.1** |
> | HR-Net-w48 | 76.3 | CA-PF | 39.8 | 81.1 |
> | HR-Net-w48 | 76.3 | Ours | **39.5** | **72.3** |
>
> - The accuracy of PandaPose remains relatively stable with respect to the errors of the 2D pose estimator. Compared with CPN and HR-Net-w48, the 2D accuracy (mAP) decrease by 7.7, but the error of PandaPose only increase by **1.5%**, while CA-PF increase by **4.5%**.
>
> - Meanwhile, as mentioned in Lines 262-263, our criterion for selecting challenging cases is based on 2D pose prediction error, i.e., these cases are the worst-performing part of 2D pose prediction in the dataset. We outperform CA-PF by a significant margin on the MPJPE in this subset (73.1 v.s. 82.4, 12.1↓). This also demonstrates our better tolerance to 2D noise.
>
> - Moreover, in Figure 9 of the main text, we gradually add Gaussian noise to the 2D pose to study the method tolerance to noise levels and our method maintains the best anti-noise ability. When the pixel-level noise scale ranges from 0 to 5, the error of PandaPose only increases by **16%** (from 39.8 to 48.2), while the image-based CA-PF increases by **56%** (from 41.4 to 64.9), and the sequence-based SOTA KTPFormer increases by **112%** (from 40.1 to 86.9). The noise resistance can be attributed to the design of anchor ensemble mechanism based on local and global anchors with ensemble joint prediction.
>
>
> In conclusion, PandaPose achieves state-of-the-art robustness to input 2D poses, which stems from our effective 3D anchor setup and anchor-to-joint mechanism.

---

> > ### Comment · Reviewer_MoUK · 2025-08-07
> > **Official Comment by Reviewer MoUK**
> >
> > I've carefully reviewed the comments of the other reviewers. I believe the weaknesses raised by reviewer 2274 are not that critical. The rebuttal solved most of my concerns, and I'll keep my rating.

---

> > > ### Author Response · Authors · 2025-08-07
> > >
> > > We sincerely thank you for your supportive and constructive feedback.  We are glad that our responses can help address your concerns, and we welcome any further questions or discussions to improve our work!

---

### Official Review · Reviewer_2274 · 2025-07-07

**Clarity:** 2
**Significance:** 2
**Originality:** 2
**Rating:** 3
**Confidence:** 4

**Summary:**

The paper proposes a method to improve the accuracy and robustness of 3D pose estimation.

The approach builds on top of 2D pose estimation and aims at
(i) overcoming limitations of 2D pose estimation accuracy
(ii) handling the issue of overlapping joints (self-occlusion) and related problems of per-joint depth estimation

For (i) the authors introduce an "adaptive anchor sampling" method, whereby they sample multiple positions around the estimated 2D joint, and use these as additional hypotheses beyond the bottom-up 2D pose estimate. Each of those anchors points provides its own 3D joint estimate, which are then aggregated into a single estimate through a convex combination - the combination weights are set by a softmax unit.

For (ii) the authors estimate a separate depth map per joint, rather than using a single depth channel. This allows every joint to receive its own supervision at training time, and at test time can provide distinct depth estimates for overlapping joints, thereby handling self occlusion.

Several ablation studies confirm the merit of the main contributions, while the results outperform the state-of-the-art on the Human3.6 dataset.

**Questions:**

Could you please provide timings of this method compared to competing baselines? This would be very relevant to interactive applications like the ones described in the introduction.

**Ethical Concerns:**

["NO or VERY MINOR ethics concerns only"]

**Final Justification:**

I remain on the fence regarding this paper because of the presentation and narrow focus of the work, but I understand that the authors are serious about their evaluation and have done their best to address the feedback of all reviewers.
I stick to my borderline reject recommendation since I still think that there is lots of space for improvement - but I would not object to the paper getting accepted if the authors do the extra work required for the camera-ready version.

**Limitations:**

Yes

**Quality:**

2

**Strengths And Weaknesses:**

Strengths:
- Some novel components are fairly validated - e.g. clear gains thanks to anchor strategy, demonstrated robustness to pose errors (also shown in Table 9).

- Interesting improvements on H3.6M compared to similar methods that require a sequence as input.


Weaknesses:
- Poor presentation. Even though I am quite knowledgeable in the field, it took me a long time to read the paper.
* The introduction section is too long. The authors attempt to cram the whole method in the intro with an intricate technical description that tries to be intuitive but is uninitelligible. For instance l. 43-45, 48-51, 57-59 are filled with technical jargon that is not explained, and the reader is left to guess what the authors mean. The introduction should be a high-level overview of the method, not a detailed description of the components.
Beyond that the text is frequently confusing, with convoluted language that does not help the reader understand the method. For instance, l. 59-61:  "For the supervision, in the absence of dense ground truth depth maps, we extract the depth value of ground truth joints instead of the complete depth as supervision."
This is a very verbose way of saying your depth loss is defined only the respective joints - and is out of place for an introduction.

Beyond confusing the reader, parts of the introduction read like propaganda, where the authors make claims that are not substantiated by the text immediately preceding or following them - for instance the text in lines 61 to 65 is impossible to validate before someone has read the full paper even then it’s unclear if this is true (maybe this is true for the second part)

* having spent two pages on an intro, then the authors spend another page on related work. They then try to cram every useful illustration or discussion in tiny figures of very low resolution, and with captions that do not explain what the reader should take away from them. For instance the  captions of Fig 11,12 do not explain what there is notice, and what the conclusion should be. Also the resolution is so low that one can hardly see what is in the image (eg MPI on 11)
Similarily, for Figure 10 an experiment is described ony in caption (not in the text), and the images are of very low resolution. It is impossible to see anything. After careful inspection it seems that for the occlusion the centroid has moved quite a bit. It is not clear if this is expected.

- Complexity: partly because of the poor presentation, but more importantly because the method is quite complicated, I find it hard to see what to take away from the paper.
There are many bells-and whistles which are introduced like systems, with little justification: e.g. in lines 224-227: "Each decoder layer comprises a depth cross-attention layer, an inter-anchor self-attention layer, and a 3D deformable cross-attention layer". This design is never validated compared to a simpler baseline, and it is impossible to see if it is really necessary and how to reproduce it.

Some other things that are fairly standard are introduced with an unwarranted level of complexity - e.g. the idea of using a separate depth map per joint. A whole section is dedicated to this (3.3), the authors mention (l. 179): "we innovatively design a joint-wise approach to predict individual depth distributions for each joint as shown in Fig 3"

while it is common practice in multi-person pose estimation  to predict per-joint depth maps or depth offsets:
SMAP: Single-Shot Multi-Person Absolute 3D Pose Estimation, ECCV 2020
3D Human Pose Estimation via Explicit Compositional Depth Maps, AAAI 2020
HEMlets Pose: Learning Part‑Centric Heatmap Triplets for Accurate 3D Human Pose Estimation” (Zhou et al., ICCV‑19)

Results:
Hard to say if the present results are practically siginificant. The main evaluation is on Human3.6M which is plagued by a very low number of subjects and background variation. The results on 3DPW are more interesting, but the authors do not compare with other methods that perform better. In particular over the past five years, several works on 3D human mesh recovery (arguably a harder problem than 3D pose estimation, that is also studied by a bigger part of the community) have reported substantially better results on 3DPW, e.g.:

HMR 2.0:  Humans in 4D: Reconstructing and Tracking Humans with Transformers
CameraHMR: Aligning People with Perspective:
TokenHMR: Advancing human mesh recovery with a tokenized pose representation

Understandably, some of them use different datasets for training. But I still find it more illuminating to use datasets and benchmarks similar to those used in the main 3D pose estimation works, rather than only look at Human3.6M results.


Minor issues:

Missing references: please see above on joint-level depth maps. Also other works that have similar ideas - e.g.
overcoming the spatial resolution of 3D voxels: Integral Human Pose Regression, ECCV 2018
Per-joint depth estimation: I2L-MeshNet, ECCV 2020

Pose prior: misnomer. There is no prior idea in the sense of prior knowledge, or prior distribution (it is conditional on the 2D joints found in the input image)

Canonical coordinate system: misnomer. For humans a canonical coordinate system would be pose invariant in 3D e.g. an A-pose skeleton. I do not see anything like that in the paper, rather the authors use a coordinate system that is defined by the 2D pose estimate, which is not canonical in any sense.

---

> ### Author Rebuttal · Authors · 2025-07-31
>
> We appreciate your careful reading of our manuscript and your comments on the writing. Actually, compared with previous methods, PandaPose has many differences in both the overall framework and specific design implementation. Due to limited space, we spend considerable effort in the introduction to help readers quickly establish the logical flow (problem - motivation - method - results), and provide further details in later sections. In the future, we will strengthen the connection between the concepts in the introduction and Figure 1, optimize the layout, and add explanatory content to eliminate misunderstandings (if accepted, there will be an additional page). Meanwhile, our writing has been generally well-received by the other four reviewers, with an average score of clarity above 3. Next, we will address all your concerns below.
> ## **Introduction writing**
> For Line 43-45, we aim to clarify that the overall architecture is built on a more advantageous initial distribution of 3D anchors and the anchor-to-joint mapping approach. This is the key innovation and main advantage of our method. Line 48-51 then detail the design of the 3D anchors. Line 57-59 and 61-65 summarize the introduction, analyze the design concepts and motivations, and explain the method's superiority in handling challenging samples, as well as its efficiency and accuracy. We expect readers to grasp important concepts such as 3D anchor space, anchor-to-joint mapping, and joint-wise depth early in the introduction.
>
> ## **LayerOut and Figure**
> Following the comparison protocol as previous works (CA-PF, MixSTE etc.), **Figure 11**,  as a qualitative comparison, highlights the parts our method has better prediction in circles as in the caption. **Figure 12** serves as an OOD experiment to tests the generlization. **Figure 10** is dedicated to demonstrating the robustness of the proposed joint-wise anchor in the case of 2D pose prediction errors as in Line 323 and the caption. All figures in paper are vector graphics and carefully designed. It is recomended to view the paper through a standard PDF reader and zoom in on the illustrations for clearer observation of details. We will enlarge some images into full-width figures in subsequent versions for better viewing.
> ## **Complexity of decoder**
> Regarding the decoder design, our approach is an improvement upon the standard deformable Transformer decoder\[1\] (comprising self-attention and deformable cross-attention). Specifically, we introduce a depth-domain cross-attention module and extend the original 2D deformable cross-attention to 3D deformable cross-attention. It is worth emphasizing that the deformable attention mechanism has been widely validated for its effectiveness across numerous computer vision tasks. Its successful application is also evidenced by pose estimation related works such as DeforHMR\[2\] (3DV 2025) and DeFormer \[3\](CVPR 2023), providing a solid foundation for our design choice.
> We further supplement the ablation experiment on the improvement of prediction by the improved components as table below. Both modules are designed to model the depth domain, and experiment proves that they can significantly enhance 3D pose prediction quality, as reflected by the decreasing MPJPE.
>
> | Depth cross-attention | Deformable attention type | MPJPE on Human3.6M |
> | --- | --- | --- |
> | no | 2D | 41.2 |
> | yes | 2D | 40.9 |
> | no | 3D | 40.3 |
> | yes | 3D | 39.8 |
>
> > [1] Deformable DETR: Deformable Transformers for End-to-End Object Detection
> >
> > [2] DeforHMR: Vision Transformer with Deformable Cross-Attention for 3D Human Mesh Recovery
> >
> > [3] Deformable Mesh Transformer for 3D Human Mesh Recovery
> ## **Joint-wise Depth Clarification**
> Overall, our approach emphasizes the necessity of accurate joint-wise absolute depth for 3D pose estimation, differing fundamentally from the methods you mentioned. SMAP[1] models root joint depths in multi-person scenarios without estimating specific joint absolute depth values, leading to potential error accumulation. The limb depth map method[2] interpolates regressed joint depths based on limb rigidity, limiting accuracy due to interpolation errors and failing in non-rigid motions. HEMlets[3] encodes only relative depth ordering without quantitative depth values, which is insufficient for precise 3D coordinate calculation.
> In contrast, our method directly estimates and utilizes joint-specific metric depths, avoiding error accumulation and capturing local depth variations caused by joint movements. Unlike SMAP's reliance on a long estimation chain prone to noise and limb depth maps' dependence on rigid limb assumptions, our approach ensures robust and high-precision 3D positioning through direct quantitative depth estimation.
>
> Moreover, the aforementioned methods lack theoretical analysis on joint-wise depth estimation. In Section 3.3, we provide a detailed theoretical and technical justification of our approach—setting us apart from previous works. Through qualitative and quantitative analysis (Figure 5, Figure 6, Table 5), we clearly articulate the motivation behind our design. Technically, our method introduces a lightweight depth encoder that predicts a joint-wise depth map with real physical values in a classification manner—distinct from existing approaches. This provides a more accurate and robust depth prior, enabling more effective feature interaction in the subsequent decoder.
> > [1] SMAP: Single-Shot Multi-Person Absolute 3D Pose Estimation
> >
> > [2] 3D Human Pose Estimation via Explicit Compositional Depth Maps
> >
> > [3] HEMlets Pose: Learning Part‑Centric Heatmap Triplets for Accurate 3D Human Pose Estimation
> ## **Result on 3DPW**
> Our experimental setup is specifically designed to evaluate cross-dataset generalization—training on Human3.6M and testing on 3DPW. This protocol is widely adopted to assess model robustness under domain shifts, such as differences in camera setups, subjects, and environments. In Table 3, all compared methods are SOTA approaches evaluated under the same cross-dataset setting, focusing on single-person 3D human pose estimation.
> We follow mainstream practices in our comparisons, and the performance metrics for all cited methods are taken directly from their publicly available results—including latest HiPART (CVPR 2025), ensuring both fairness and up-to-date benchmarking.
> We emphasize that our goal is not to establish new absolute performance records on 3DPW spanning different settings and methodological paradigms, but rather to evaluate how well models generalize from controlled, lab-based settings (Human3.6M) to more challenging, in-the-wild scenarios (3DPW). This evaluation paradigm is highly relevant in real-world applications, especially when labeled in-the-wild data is limited or expensive to acquire.
>
> It is worth noting that many mesh-based methods achieving strong performance on 3DPW are trained on large, mixed datasets that include in-the-wild or synthetic data. A direct comparison under such conditions would be unfair. To enable a more equitable comparison with mesh-based approaches, we retrain TokenHMR (CVPR 2024) under our same experimental setup. Under this setting, TokenHMR achieves an error metric MPJPE of 78.5 on 3DPW, which is higher than our result of 74.9, further demonstrating the effectiveness of our method in this generalization scenario.
> ## **Clarification of Human3.6M**
> In response to the reviewer's doubts about whether the results on Human3.6M are practically significant, we would like to clarify that Human3.6M remains a widely recognized standardized benchmark in the field of 3D pose estimation, with advantages in high-quality and controllable annotations. Recently published papers still use this dataset as a core benchmark ( HiPART[CVPR25], UPose3D[ECCV24] etc.).
> We also observe that the metrics of current 3D human mesh methods on Human3.6M are often inferior to those of 3D human pose estimation (as shown in the table below).
>
> More importantly, although Human3.6M is a relatively "clean" dataset, we observe that existing pose methods still perform poorly in occlusion scenarios within this dataset. For instance, we select complex scenarios in the dataset by measuring 2D pose error > 5(line 262-263 in main text), which often represent self-occlusion and motion blur. Existing methods often lose information about key body parts, leading to a decline in overall pose estimation accuracy. This observation is precisely the motivation and core contribution of our work. Our method demonstrates significant performance improvements in these challenging subsets. For instance, on challenging subset (~5% in the full test set), the MPJPE of ours is 9.3 mm lower than that of the current SOTA method CA-PF (73.1 vs 82.4) as shown in Table 1 of main text.
>
> | Method | MPJPE on Human3.6M |
> | --- | --- |
> | ProHMR (Mesh-based)| 65.1 |
> | HMR2.0  (Mesh-based)| 52.8 |
> | ScoreHMR  (Mesh-based)| 47.9 |
> | Ours | 39.8 |
> ## **Minor issues**
> 1. Pose prior：In our paper, we adopt the term "Pose prior" following the convention of previous related works (e.g., C3P[1] [ECCV2022], DiffPose[2] [CVPR2023]), which also used this expression in similar contexts. In these works, "prior" does not strictly refer to prior knowledge or distribution in the Bayesian sense, but more broadly refers to the initial pose hypothesis derived from the 2D joint positions in the input image. This usage aims to emphasize the process by which the model uses the known 2D information as a basis to predict 3D poses. We will also strive to supplement further background in the revised version to eliminate ambiguity.
>
> 2. Canonical coordinate system：The canonical coordinate system in the text refers to the 3D world coordinate system with actual physical values, rather than the 2D pose coordinate system. The 3D anchors are located in the same coordinate system with 3D human pose. We will also further strengthen this concept in the paper.

---

> > ### Author Response · Authors · 2025-08-05
> > **Further detailed ablation on decoder**
> >
> > The reviewer G3SQ read through the comments of other reviewers and our response, and believes that most concerns have been addressed, except that the ablation study on the decoder could be more detailed. In our initial response above regarding **complexity of decoder**, we focus on justifying the enhancements we introduce to the standard deformable transformer decoder ( i.e., the depth cross-attention and 3D deformable attention)—designed for depth-aware modeling.  To more clearly and thoroughly illustrate the individual contributions of the three attention modules, we further conduct ablation studies starting from the simplest baseline. The expanded results on Human3.6M are provided in the table below.
> >
> > | Row index | Depth cross-attention | Anchor self-attention | Deformable cross-attention | MPJPE (full) | MPJPE (challenging) |
> > | --- | --- | --- | --- | --- | --- |
> > | 1 | \- | \- | \- | 44.9 | 86.8 |
> > | 2 | \- | $\\checkmark$ | \- | 42.7 | 84.7 |
> > | 3 | \- | $\\checkmark$ | 2D | 40.9 | 80.8 |
> > | 4 | $\\checkmark$ | \- | 2D | 41.2 | 80.3 |
> > | 5 | $\\checkmark$ | $\\checkmark$ | 2D | 40.7 | 79.5 |
> > | 6 | $\\checkmark$ | \- | 3D | 40.4 | 76.0 |
> > | 7 | \- | $\\checkmark$ | 3D | 40.3 | 76.4 |
> > | 8 | $\\checkmark$ | $\\checkmark$ | 3D | **39.8** | **73.1** |
> >
> > The method in Row 1 refers to directly predicting the anchor-to-joint offset/weight from the input $Q\_{anchor}$ . It can be seen that all three attention modules directly improve prediction performance. Here we highlight some impressive points:
> >
> > - From row 2 - row 3, the 2D deformable cross-attention allows anchor features to interact with visual features, which significantly enhances the semantic nature of anchors and prediction performance (42.7 - 40.9).
> >
> > - From row 6 - row 8, the introduction of anchor self-attention helps to establish the connection in anchor self articulation, while also exerting a positive impact on performance (40.4 - 39.8).
> >
> > - From row 7- row 8, depth cross-attention explicitly equips anchor features with depth domain characteristics, resulting in performance improvement (40.3 - 39.8). Notably, particularly in challenging cases, notable enhancements are observed (76.4 - 73.1).
> >
> > - From row 5 - row 8, the 2D deformable attention is replaced with 3D deformable attention. The performance is further improved (40.7 - 39.8) due to the direct addition of depth distribution for depth-visual-anchor feature interaction. Especially in challenging cases, the introduction of depth features is very helpful in alleviating depth ambiguity and self-occlusion (79.5 - 73.1).
> >
> >
> > These also align with the conclusions presented in Table 5, Line 311-319 of the main text.
> >
> > We sincerely hope that our response can address your concerns. If you have any more concerns or questions, we are more than willing to assist in addressing them!

---

> > > ### Comment · Reviewer_2274 · 2025-08-07
> > >
> > > Dear authors- I think you should place the text above (Further detailed ablation on decoder) in your reply to "reviewer G3SQ" rather than me - there is a change the reviewer will not see it.

---

> > > > ### Author Response · Authors · 2025-08-08
> > > >
> > > > Thanks for your feedback. Here are some clarifications:
> > > > 1. The comment above provides an expanded explanation regarding your Q3 (i.e., the complexity of the decoder). At the request of Reviewer G3sQ, we conduct additional detailed ablations  and share these results with you as well. Reviewer G3sQ, after reviewing all the reviews and rebuttals during the discussion phase, noted: “most of the issues have been resolved, except for: (Reviewer 2274) More ablation study about 3D anchor interaction decoder.” We provide further clarification in response to G3sQ’s comment, fully addressing their concern, and share this additional analysis with you to further clarify any remaining questions.
> > > > 2. Our official rebuttal is submitted above. The additional comment is solely a supplementary response to the discussion initiated by Reviewer G3sQ. If you feel this supplementary content is not directly relevant to your original review, please feel free to disregard it. We receive and fully understand the notification from the Program Committee—this additional ablation study is part of the open discussion, while our official rebuttal is submitted on time and  targeted to address all raised concerns.
> > > > 3. Should you have any further questions regarding our official rebuttal, we welcome continued discussion and are happy to provide additional clarification.

---

### Note · Authors · 2025-08-12

Dear Area Chairs and Reviewers:

We sincerely appreciate your time and efforts for the meticulous reviews and insightful comments on our paper.

We are glad to know that the reviewers have recognized the strengths of our work.
Specifically, Reviewer MoUK highlights the **innovative methodology** and **superior performance** of our approach, while Reviewer G3sQ commends its **excellent performance** and the **clarity of writing**. Reviewer HbVK further acknowledges that our method is **clearly written**, features a **well-motivated design**, and achieves **strong empirical results**. Additionally, Reviewer Cg4S emphasizes the **robustness and generalization capability** of our framework.  We are encouraged by the reviewers’ positive feedback on multiple aspects of our work, including the methodological design, experimental evaluation, and clarity of presentation. Their recognition of the framework’s novelty, effectiveness, and generalizability is greatly appreciated.

Meanwhile, We carefully address the questions raised by the reviewers and provide detailed responses to each comment. We are grateful for the positive feedback from Reviewer Cg4S, MoUK and G3sQ. For instance, Reviewer MoUK and G3sQ  read through all the reviewers' comments and conclude that "most issues have been resolved." Reviewer MoUK also believes that the weaknesses proposed by Reviewer 2274 are not critical.

Besides, during discussion period, we conduct additional detailed ablation studies on the 3D anchor interaction decoder in response to  Reviewer G3sQ's suggestion, who recommend expanding upon the initial ablation analysis requested earlier by Reviewer 2274.  We provide further clarification in response to G3sQ’s comment,  and share this additional analysis with Reviewer 2274 to further clarify any remaining questions.
We hope our rebuttal offers clear and thorough explanations to resolve all raised issues.

Thanks again for your time and efforts to review our paper.

---

### Decision · Program_Chairs · 2025-09-17

**Decision:**

Accept (poster)

**Comment:**

Four of the five reviewers recommend acceptance. The fifth point out that the paper lacks clarity. I urge the authors to take his comments to heart and to improve the writing and organization when preparing the final version.